# Perturbed structural dynamics underlie inhibition and altered efflux of the multidrug resistance pump AcrB

Eamonn Reading [1,5,6✉], Zainab Ahdash[1,5], Chiara Fais[2], Vito Ricci[3], Xuan Wang-Kan [3], Elizabeth Grimsey[3], Jack Stone[3], Giuliano Malloci [2], Andy M. Lau[1], Heather Findlay[1], Albert Konijnenberg[4], Paula J. Booth[1], Paolo Ruggerone [2], Attilio V. Vargiu [2], Laura J. V. Piddock [3] & Argyris Politis [1,6✉]

Resistance–nodulation–division efflux pumps play a key role in inherent and evolved multi-drug resistance in bacteria. AcrB, a prototypical member of this protein family, extrudes a wide range of antimicrobial agents out of bacteria. Although high-resolution structures exist for AcrB, its conformational fluctuations and their putative role in function are largely unknown. Here, we determine these structural dynamics in the presence of substrates using hydrogen/deuterium exchange mass spectrometry, complemented by molecular dynamics simulations, and bacterial susceptibility studies. We show that an efflux pump inhibitor potentiates antibiotic activity by restraining drug-binding pocket dynamics, rather than preventing antibiotic binding. We also reveal that a drug-binding pocket substitution discovered within a multidrug resistant clinical isolate modifies the plasticity of the transport pathway, which could explain its altered substrate efflux. Our results provide insight into the molecular mechanism of drug export and inhibition of a major multidrug efflux pump and the directive role of its dynamics.

[1] Department of Chemistry, King's College London, Britannia House, 7 Trinity Street, London SE1 1DB, UK. [2] Department of Physics, University of Cagliari, Cittadella Universitaria, S.P. Monserrato-Sestu, 09042 Monserrato, (CA), Italy. [3] Antimicrobials Research Group, Institute of Microbiology and Infection, College of Medical and Dental Sciences, The University of Birmingham, Birmingham B15 2TT, UK. [4] Thermo Fisher Scientific, Zwaanstraat 31G/H, 5651 CA Eindhoven, The Netherlands. [5] These authors contributed equally: Eamonn Reading, Zainab Ahdash. [6] These authors jointly supervised this work: Eamonn Reading, Argyris Politis. ✉email: eamonn.reading@kcl.ac.uk; argyris.politis@kcl.ac.uk

AcrB is a homotrimeric integral membrane protein that forms part of the tripartite AcrAB-TolC efflux pump[1] (Fig. 1a, b and Supplementary Fig. 1). Energized by the proton-motive force, AcrB transports a broad variety of toxic substances, including antibiotics, outside of the cell through a channel formed by the periplasmic adaptor protein, AcrA, and outer membrane channel, TolC[2,3]. It is constitutively expressed in many pathogenic Gram-negative bacteria and, with its homologues forming the most clinically relevant pumps, has become a target for drug discovery to tackle multidrug resistance (MDR)[4–7].

Previous structural and biochemical work has enabled drug-binding pockets and efflux pathways for AcrB to be proposed[1,2,7–17]. Drug transport is purported to occur via cooperative rotation between three distinct monomer conformations: loose (L), tight (T), and open (O) (Fig. 1c). Where, in the L-state, drugs gain access to the proximal binding pocket (PBP) through entrance channels[1,8,18]. Upon a conformational change to the T-state, the drug is then moved towards the distal binding pocket (DBP) before being transported through the exit channel of the periplasmic domain, following a second conformational change from the T- to O-state[19,20]. A switch-loop (615FGFAGR620) acts to separate the PBP from the DBP (Fig. 1b); the conformational flexibility of this region is proposed to be essential for the regulation of substrate binding and export[21,22].

Each AcrB monomer contains a connecting-loop, which protrudes into the Funnel Domain (DC and DN subdomains) of the adjacent monomer (Fig. 1a) and has been shown to be important for stabilizing the trimer during the functional rotation[23]. During rotation, the transmembrane (TM) region—which consists of the R1 (TM1-6) and R2 (TM7-12) domains connected by the Iα-helix (520FEKSTHHYTDSVGGIL535) (Fig. 1a, b)—is also postulated to move considerably[14,24].

Despite the availability of high-resolution structures of AcrB, it has remained challenging to dissect how its molecular mechanisms are regulated by naturally occurring MDR mutations, efflux pump inhibition and their synergy with antibiotics. One fundamental aspect of its structure that remains unresolved concerns the role of its structural dynamics, which are often crucial for protein function[25–28]. Here, we use hydrogen/deuterium exchange mass spectrometry (HDX-MS) to directly measure changes in AcrB structural dynamics owing to binding of the ciprofloxacin (CIP) antibiotic and the well-studied phenylalanine-arginine-β-naphthylamide (PAβN) efflux pump inhibitor (EPI)[5,7,15,29]. Investigating both wildtype AcrB (AcrB^WT) and a recently discovered G288D mutation (AcrB^G288D), uncovered in a post-therapy MDR clinical isolate of *Salmonella* Typhimurium[30], we were able to further understand the structural and functional consequences of substrate and inhibitor binding and clinically relevant mutation.

In this work, we reveal that the PAβN EPI restricts the intrinsic motions of the drug-binding pockets as part of its mechanism of action and is effective against both AcrB^WT and AcrB^G288D. We also demonstrate that an EPI can dually bind

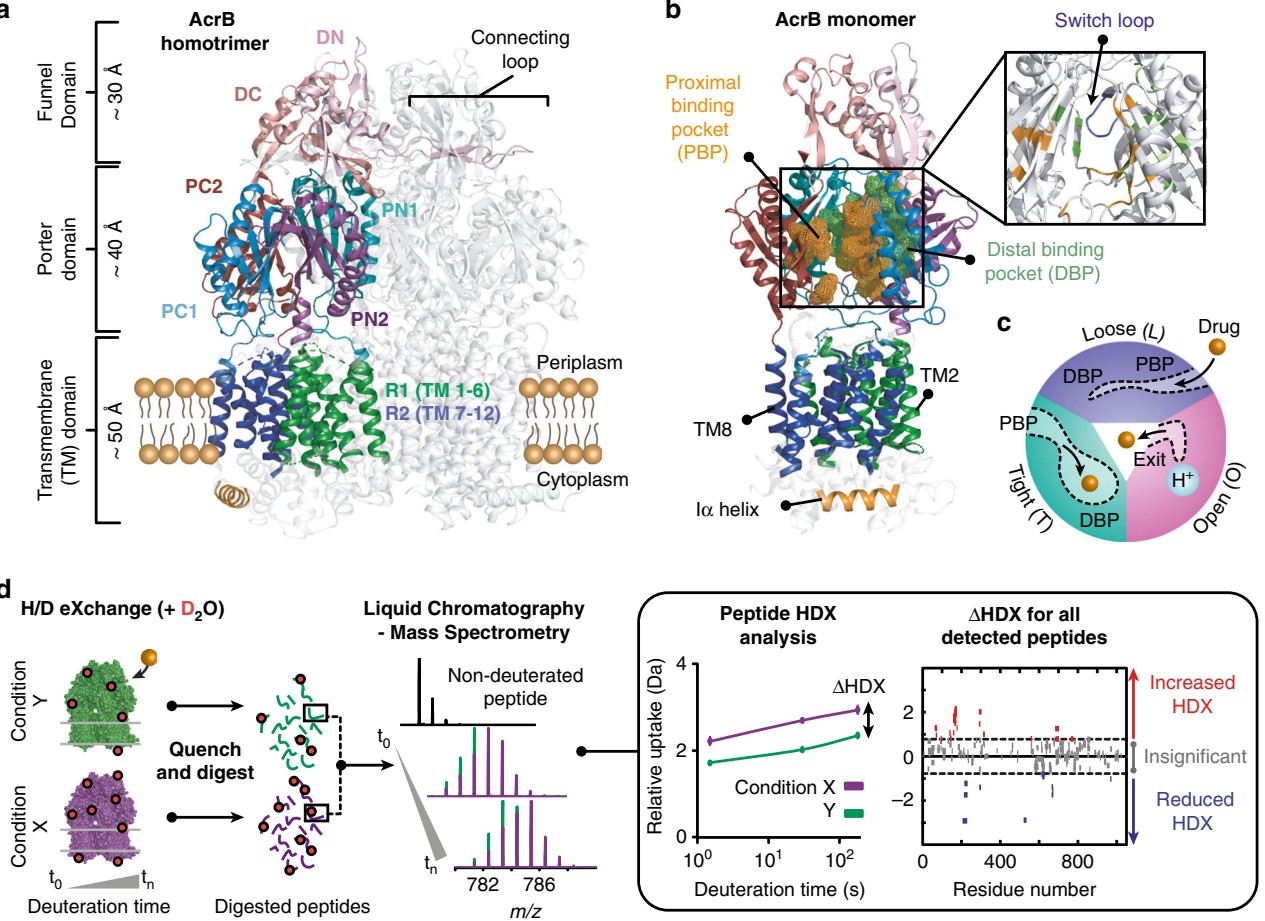

**Fig. 1 Summary of structure, function, and HDX-MS of AcrB. a** Structure and subdomains of AcrB (PDB:2HRT). **b** Highlighted regions of functional interest on AcrB monomeric unit. **c** Functional rotation drug efflux mechanism between the three distinct monomer conformations within trimeric AcrB. **d** HDX-MS workflow.

to AcrB alongside an antibiotic, without affecting its inhibitory action. We discover that an MDR mutation in *acrB* impacts upon the structural dynamics of the efflux translocation pathway, likely contributing to its modified substrate efflux. Structural dynamics therefore have a critical role in the inhibition and substrate efflux of AcrB. Understanding the effect of these dynamics on structure and function is critical for successful assessment of resistance–nodulation–division (RND) efflux pump mechanism and inhibition, especially in relation to MDR-conferring mutations.

## Results

**HDX-MS of AcrB.** HDX-MS is a solution-based method which can provide molecular-level information on local protein structure, stability, and dynamics[31–33]. HDX occurs when backbone amide hydrogens are made accessible to $D_2O$ solvent through structure unfolding and H-bond breakage; HDX is fast within unfolded regions and slow within stably folded regions (i.e. α-helices, β-sheet interiors), where local structural fluctuations which expose an otherwise protected amide hydrogen to solvent transiently are required for HDX to occur. In order to decipher the effect of drug binding and mutation on AcrB structural dynamics, we performed differential HDX (ΔHDX) analysis between two conditions (e.g. drug-bound and drug-free), which is a sensitive approach for detecting associated structural perturbations between two different protein states (Fig. 1d)[33–35].

We optimized HDX-MS conditions on *Escherichia coli* (*E. coli*) AcrB solubilized within *n*-Dodecyl-β-D-Maltopyranoside (DDM) detergent micelles achieving 72% peptide coverage (Supplementary Fig. 2). A relative fractional deuterium uptake analysis of AcrB revealed that many of its residues form part of stable structures, inferred from the longest labelling time points (0.5–1 h) required for substantial deuterium incorporation[32] (Supplementary Figs. 3, 4). Its most structurally dynamic regions being discovered within the subdomains of the Porter Domain (PC1, PC2, PN1, and PN2) which, notably, host the main drug-binding pockets (Fig. 1a, b and Supplementary Table 1). Lack of extensive HDX was observed within the TM domains, likely afforded by their protection within the hydrophobic environment of the detergent micelle.

**EPI restricts drug-binding pocket dynamics.** First, we examined how CIP and PAβN binding affects AcrB[WT] structural dynamics. CIP is a licensed antibiotic in clinical use and has been demonstrated to bind to the DBP, PC1/PC2 cleft and central cavity by MD simulations and X-ray crystallography[15,29]. While PAβN, an EPI and substrate of AcrB, has been shown to bind to similar areas of AcrB as many antibiotics, this binding may be at distinct sites[5,15,29].

In the presence of CIP only a few regions within the PN2 subdomain, central cavity, R2 domain (TM 7-12) and Iα-helix demonstrated significant differential HDX (Fig. 2a, b), suggesting that CIP binding only subtly alters the structural motions of AcrB[WT]. The presence of PAβN gave comparably increased HDX within the PN2 subdomain. However, in stark contrast to CIP, inhibitor binding led to HDX reduction throughout extensive parts of the PC1/PC2 cleft of the drug-binding pockets and within the connecting-loop (Fig. 2a, b and Supplementary Fig. 5). This could signify inhibitor-induced structural stabilization of the drug-binding pocket entrances.

HDX of the Iα-helix did not significantly change upon addition of PAβN, whereas HDX was substantially increased upon CIP binding (Fig. 2a), possibly because the inhibitor weakens the coupling between the R1 and R2 TM domains. Notably, switch-

loop spanning peptides with reduced HDX were detected when PAβN was present, which may reflect a binding interaction and/or structural stabilization (Fig. 2a, b and Supplementary Fig. 5).

Together, our HDX data support an inhibitory mode of action by which an EPI primarily acts to impart concerted restraint on AcrB structural dynamics, notably restricting the drug-binding pockets, connecting- and switch-loops.

Multi-copy 1-μs long MD simulations of AcrB[WT] bound to PAβN confirmed that—in agreement with the previous literature[6]—the inhibitor could stably bind to the DBP, straddling the switch-loop and establishing strong interactions with the hydrophobic trap (HT); a peculiar region of the DBP rich in phenylalanines and involved in EPI binding[4,36] (Supplementary Fig. 6).

To further understand the dynamics of AcrB in the presence of substrates, Root Mean Square Fluctuation (RMSF) and first hydration shell profiles were compared to the HDX-MS data. The average number of water molecules in the first amide NH solvation shell computed by MD simulations has been found to correlate well with HDX[37]. A reduced hydration shell should therefore imply reduced HDX, due to the decrease in specific interactions between amide N–H bonds and the solvent. However, protein HDX is complex, with neighbouring residues having significant differences in their solvent interactions, this combined with the stark contrast between MD simulation and HDX-MS experimental time scales (μs to ms versus seconds to hours) means that a simple quantitative comparison can often be incomplete. Nevertheless, comparisons to MD calculated hydration profiles can provide informative qualitative interpretation of protein HDX.

MD analysis of AcrB[WT]-PAβN (T monomer) and apo AcrB[WT] (L monomer) revealed that binding of PAβN is accompanied by an overall rigidification of the protein (Supplementary Fig. 7; see also Supplementary Discussion and Supplementary Tables 2, 3), which involves large patches of the DBP, PBP, switch-loop, as well as the exit channel gate (EG), CH1, and CH2 channels[8]. In particular: (i) regions containing residues belonging/adjacent to the switch-loop that were found to directly interact with PAβN become more rigid in its presence, the extent of HDX protection upon PAβN binding (as revealed by the HDX-MS data) correlating with the formation of hydrogen bonds between the EPI and residues of (and nearby) the DBP; ii) the switch-loop itself (residues 615–620) features moderately enhanced hydration, whereas the nearby segments (residues 612–614 and 621–624) are overall dehydrated with respect to apo AcrB[WT]. This supports an interaction between PAβN and the switch-loop region, which we anticipate being a key factor in mediating the mode of action of this EPI. More generally, the structural stabilization that occurs upon PAβN binding might prevent local, as well as distal, functional movements that are key to substrate efflux along the transport pathway.

Overall, these data agree with a model for inhibitor action, which has been proposed to work by trapping AcrB in a conformation, possibly a T-like state, which prevents adequate functional rotation and substrate transport[16].

**EPI and antibiotic can dually bind to AcrB.** To fully understand EPI action, it is essential to consider its activity in the presence of antibiotic substrates, especially considering the emerging importance of drug combination therapies for treating bacterial infection[38]. Antibiotic susceptibility assays against *E. coli* confirmed the ability of the PAβN EPI to potentiate antibiotic activity (Fig. 2c). PAβN increased antibiotic susceptibility for a range of antimicrobial AcrB substrates (ciprofloxacin (CIP), tetracycline (TET), and chloramphenicol (CHL)) in a substrate-dependent

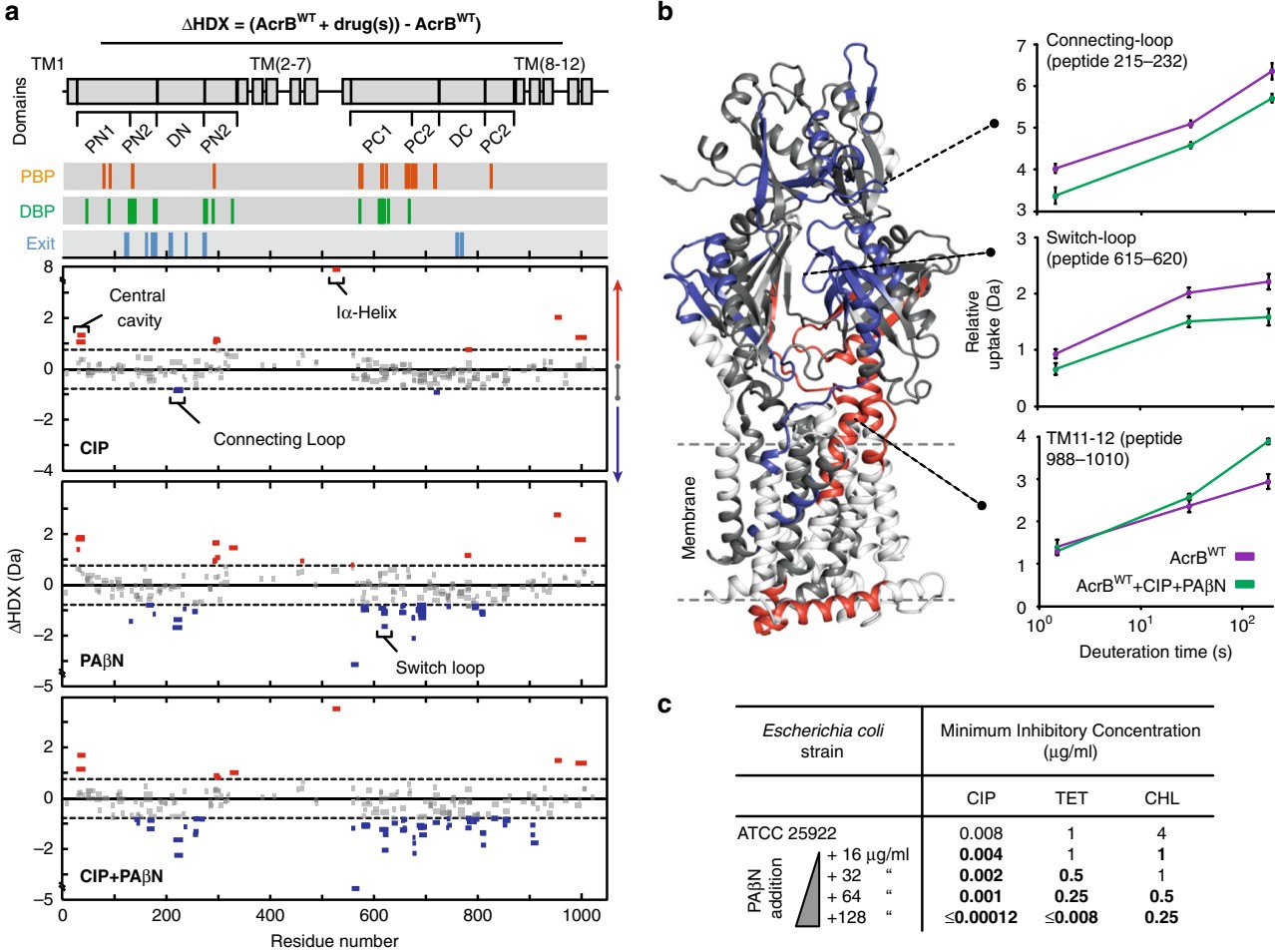

**Fig. 2 Effect of EPI PAβN on AcrB structural dynamics and function. a** Sum differential HDX (ΔHDX) plots for different drug conditions (ΔHDX = (AcrB$^{WT}$+drug(s))−AcrB$^{WT}$) for all time points collected. Red signifies peptides with increased HDX between states and blue represents peptides with decreased HDX. 98% confidence intervals are shown as grey dotted lines and grey data are peptides with insignificant ΔHDX. All measurements were performed at least in triplicate. All supporting HDX-MS peptide data can be found in the Source Data file. **b** ΔHDX extent for (AcrB$^{WT}$+CIP + PAβN) − AcrB$^{WT}$ is coloured onto the L-state monomer of AcrB (PDB:2HRT) using Deuteros[34]. Connecting-loop from the adjacent monomer is included. Uptake plot data are the average and standard deviation from repeated measurements (n = 3). **c** MIC assays of *Escherichia coli* in the presence of inhibitor and antibiotics. Values that demonstrate >2-fold reduction in MIC are in bold. Ciprofloxacin = CIP, Tetracycline = TET, Chloramphenicol = CHL, and phenylalanine-arginine-β-naphthylamide = PAβN. MIC for PAβN alone is 256 μg/ml.

manner, with better effectiveness observed at lower PAβN concentrations for CIP than for TET and CHL.

To explore the effect of an antibiotic on efflux inhibition we performed HDX-MS in the presence of both CIP and PAβN (AcrB$^{WT}$-CIP-PAβN). We anticipated that the presence of equimolar CIP may interfere with PAβN binding, thereby affecting its ability to prevent functionality of the transporter through dynamic restrain. This was not the case. The presence of CIP did not alter the action of the PAβN inhibitor, as revealed by the strikingly similar differential HDX profiles for both AcrB$^{WT}$-PAβN and AcrB$^{WT}$-CIP-PAβN (Fig. 2a, b and Supplementary Fig. 5). Consequently, we investigated the possibility that PAβN acts by outcompeting CIP binding to AcrB. To test this, we exploited the innate fluorescence of CIP to perform fluorescence polarization binding and competition assays[39]. Interestingly, we found that CIP binds with comparable affinity to both AcrB$^{WT}$ ($K_D$ of 74.1 ± 2.6 μM from Su et al.[39]) and a preformed AcrB$^{WT}$–PAβN complex ($K_D$ of 67.3 ± 13.2 μM) (Fig. 3a), and that titration of PAβN EPI could not effectively outcompete CIP binding from a AcrB$^{WT}$–CIP complex (Fig. 3b). These

data suggest that antibiotic and inhibitor may be able to simultaneously bind at different (sub)sites within the voluminous DBP.

To further support this hypothesis, we performed blind docking calculations and MD simulations on AcrB$^{WT}$–PAβN–CIP (Supplementary Figs. 8, 9). Importantly, both drugs stably bind to the DBP within the T-state monomer, with PAβN partly occupying the HT and CIP lying in proximity of the PBP/DBP interface (Fig. 3c). The simultaneous binding of CIP and PAβN has similar effects on the flexibility and hydration of the DBP as the binding of PAβN only (Fig. 2a, b and Supplementary Fig. 10; see Supplementary Discussion). Several interactions contribute to stabilize this configuration, including the formation of hydrogen bonds between the substrates and several residues of the DBP (Supplementary Table 4) and stable intermolecular hydrogen bonds between the two ligands (Supplementary Fig. 9).

Overall, our data support the hypothesis that PAβN does not compete or prevent antibiotic binding (competitive inhibition). Instead, we propose that it inhibits AcrB function by enforcing a more restrained state, thus, reducing the frequency and

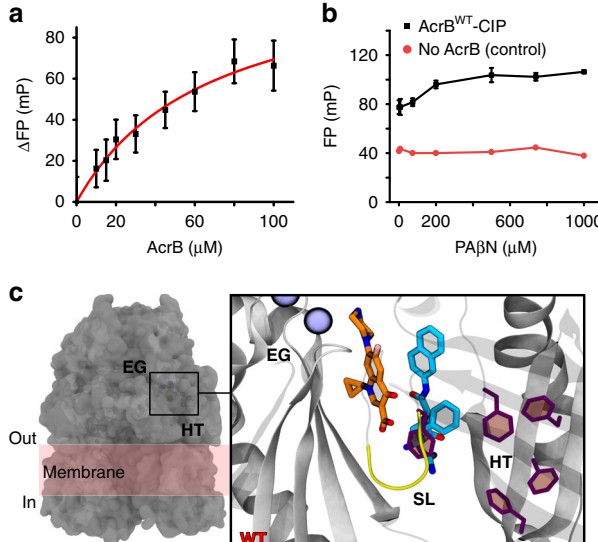

**Fig. 3 Dual binding of ciprofloxacin antibiotic and PAβN inhibitor to the DBP of AcrB$^{WT}$. a** Binding of CIP by AcrB$^{WT}$ in the presence of 150 μM of PAβN as determined by a fluorescence polarization assay performed by Su et al.[39]. CIP was maintained at 1.5 μM throughout and its emission wavelength measured at 415 nm. Reported data are the average and standard deviation from independent measurements ($n = 3$) and were fitted to a hyperbola function (FP = ($B_{max}$*[protein])/($K_D$ + [protein]), $R^2 = 0.98$). **b** Binding competition assay between PAβN and CIP for AcrB$^{WT}$. PAβN was non-fluorescent in the experimental conditions. Data are the average and standard deviation from independent measurements ($n = 3$). **c** Molecular docking and multi-copy μs-long MD simulations reveal stable interactions of CIP (orange) and PAβN (cyan) to AcrB$^{WT}$ T-state monomer and show their likely binding locations. EG = exit channel gate (blue spheres), SL = switch-loop (yellow), and HT = hydrophobic trap (purple). All computational data can be found in Supplementary Table 2 and Supplementary Figs. 8, 9.

magnitude of the conformational changes within the substrate translocation path. Its effectiveness being substrate dependent.

**MDR-conferring G288D mutation affects drug-binding pocket dynamics.** We next turned our attention to the substitution mutation, G288D (AcrB$^{G288D}$), which was found to cause resistance to some drugs (e.g. CIP) in *Salmonella*, but susceptibility to others (e.g. minocycline (MIN))[30]. G288 is a highly conserved residue in *Enterobacteriaceae*[30], suggesting an important structural role, and is present aside the HT within the DBP (Fig. 4a). We found that the G288D substitution does not alter the oligomeric state, average secondary structure content or thermal stability of AcrB, as judged by native mass spectrometry and circular dichroism (Supplementary Fig. 2). Yet, our HDX analysis revealed that the G288D mutation has a noticeable effect on the structural dynamics of AcrB. We observed that, when no drugs were present, the G288D mutation caused increased HDX for several peptides spanning the PN2 region of the protein, but decreased HDX within the PC1/PC2 regions and the connecting-loop (Fig. 4b). Seemingly, this single-point mutation can cause a long-range change to the structural dynamics of AcrB, possibly reflecting global conformational changes in its substrate-free state.

Upon substrate binding, PAβN caused reduced HDX within the PC1/PC2 regions for AcrB$^{G288D}$ (Supplementary Fig. 11), as was observed for AcrB$^{WT}$ (Fig. 2a), with AcrB$^{G288D}$-PAβN having increased HDX reduction within PC1 and R2 (TM 7-12) domains in comparison to AcrB$^{WT}$-PAβN (Fig. 4b). This

supports that the PAβN EPI affects the dynamics of the different AcrB genotypes in a similar manner, with AcrB$^{G288D}$–PAβN possibly undergoing further restraint than AcrB$^{WT}$–PAβN. Whereas CIP caused increased HDX throughout extensive regions of AcrB$^{G288D}$ (Supplementary Fig. 11), bringing the PC1/PC2 and DC regions of AcrB$^{G288D}$ closer in parity with AcrB$^{WT}$ for both CIP and CIP–PAβN conditions (Fig. 4b).

Markedly, in the *apo* form and for all three substrate conditions tested (CIP, PAβN, and CIP–PAβN), the G288D substitution consistently caused increased HDX within the PN2 region and decreased HDX of the connecting-loop (Fig. 4b). Signifying that these effects are retained even upon substrate binding and, due to their close structural proximity, may relate to concerted changes to the dynamics of the substrate translocation pathway (Fig. 4c).

**PAβN EPI inhibits both wildtype and G288D AcrB.** MD simulations of AcrB$^{G288D}$ in the presence of PAβN were performed to better understand how the G288D mutation affects EPI interactions with the drug-binding pockets. PAβN was found to bind to the HT of AcrB$^{G288D}$, interacting with the mutated D288 residue through the formation of direct and water-mediated hydrogen bonds (Supplementary Fig. 12 and Supplementary Table 5). Interactions with the aromatic residues of the HT involve hydrophobic stacking as well as cation-π attraction, not observed in AcrB$^{WT}$–PAβN (Supplementary Table 6; see Supplementary Discussion), and possibly promoted by the direct interaction of the inhibitor with residue D288 (Supplementary Table 6; see Supplementary Discussion). This interaction is supported by the reduced HDX found within D288 containing peptides of AcrB$^{G288D}$–PAβN (Supplementary Fig. 11b). Moreover, in accordance with HDX-MS, the switch-loop and the surrounding DBP region undergo further dehydration in AcrB$^{G288D}$-PAβN (Supplementary Fig. 13) and are involved in high-occurrence interactions with PAβN, similar to those formed within AcrB$^{WT}$–PAβN (Supplementary Tables 3, 6). These data, together with the direct interactions detected between the EPI and regions of the switch-loop, support the hypothesis that stabilization of the latter has a role for inhibitor mode of action of PAβN both in AcrB$^{WT}$ and in AcrB$^{G288D}$.

Similar conclusions emerged from the comparison between AcrB$^{G288D}$–CIP–PAβN and AcrB$^{WT}$–CIP–PAβN. Indeed, MD simulations of the former complex revealed that, even upon G288D substitution, CIP and PAβN can stably occupy the DBP at the same time (Fig. 5a). As in AcrB$^{WT}$–CIP–PAβN, stabilizing interactions include several substrate contacts with the AcrB$^{G288D}$ protein, also involving D288 (Supplementary Figs. 14, 15 and Supplementary Table 7), as well as intermolecular hydrogen bonds between the two ligands (Supplementary Table 8).

Fluorescence polarization binding and competition assays support that a ternary AcrB$^{G288D}$–CIP–PAβN is also possible (Fig. 5b, c): (i) CIP binds to a preformed AcrB$^{G288D}$–PAβN complex ($K_D$ of $22.7 ± 2.9$ μM) with similar, albeit slightly higher, affinity compared to CIP binding to AcrB$^{WT}$–PAβN ($K_D$ of $67.3 ± 13.2$ μM); (ii) titration of the PAβN inhibitor could not effectively outcompete CIP binding from AcrB$^{G288D}$–CIP, as was found for AcrB$^{WT}$–CIP (Fig. 3b). Taken together, the fluorescence polarization binding assays and MD simulation data advocate that AcrB$^{G288D}$ is inhibited by PAβN in a similar manner as AcrB$^{WT}$.

These findings were supported by bacterial susceptibility assays on *E. coli* containing overexpressed AcrB$^{G288D}$. AcrB$^{G288D}$ was previously discovered within *Salmonella* clinical isolates[30] and found to have increased and decreased susceptibility to MIN and CIP antibiotics, respectively. We chose, therefore, to study these AcrB substrates in the presence of the inhibitor PAβN to observe

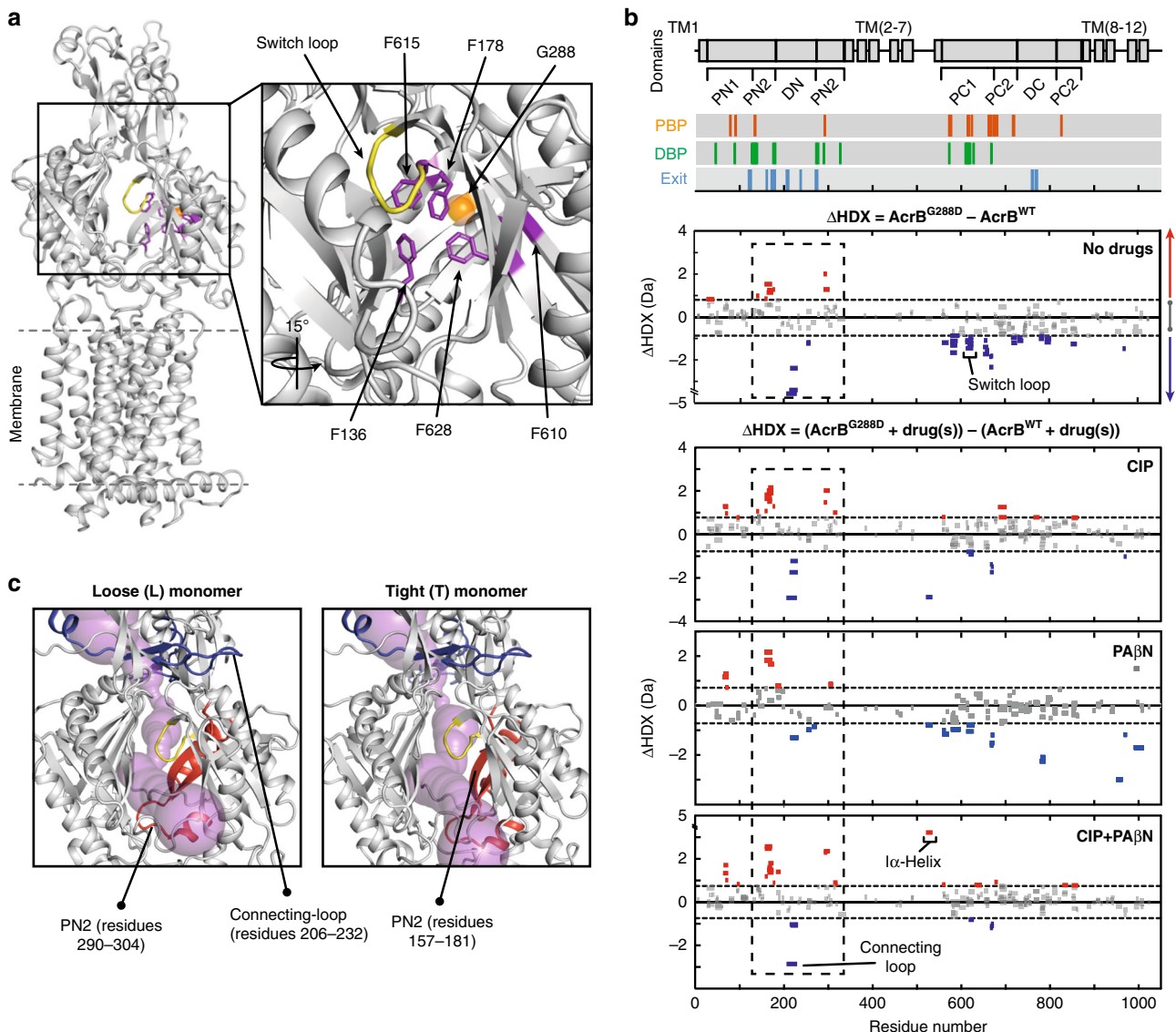

**Fig. 4 Influence of MDR mutation G288D on AcrB structural dynamics. a** In this zoom-in view of the DBP, within the L-state monomer of AcrB (PDB:2HRT), the G288 position is shown as an orange sphere amongst the hydrophobic trap (purple) and alongside the switch loop (yellow). **b** Differential HDX (ΔHDX) plots for AcrB[G288D] and AcrB[WT] with and without drugs. All data reported as in Fig. 2. 98% confidence intervals are shown as grey dotted lines and grey data are peptides with insignificant ΔHDX. All measurements were performed at least in triplicate. All supporting HDX-MS peptide data can be found in the Source Data file. (**c**) Shared regions surrounding the DBP whose HDX are increased (red) and decreased (blue) in a substrate-independent manner (highlighted by the dashed boxes in Fig. 4b) are shown on both L- and T-state monomers of AcrB (PDB:2HRT). The connecting-loop from the adjacent monomer is included. Substrate efflux pathways (magenta) for the periplasmic entrance are depicted using CAVER[68].

what, if any, effect would be seen with the different AcrB genotypes. PAβN incubation led to increased MIN and CIP antibiotic susceptibility for both AcrB[WT] and AcrB[G288D] (Fig. 5d). AcrB[G288D] being more susceptible to PAβN than AcrB[WT]. The decreased susceptibility of AcrB[G288D] to CIP, found in *Salmonella*[30], was not recapitulated in our assays using the laboratory *E. coli* strain MG1655. This may be due to CIP efflux via another transporter found in *E. coli* but not in *Salmonella*. However, the associated increased susceptibility to MIN was observed (Fig. 5d), supporting that G288D has a profound impact on AcrB substrate efflux within both *E. coli* and *Salmonella*.

## Discussion

In summary, we found that binding of an EPI, PAβN, restricts AcrB dynamics and could not be outcompeted by an antibiotic, CIP, whose activity it potentiates. Fluorescence binding, MD

simulations, and docking studies supporting the existence of a ternary protein–EPI–antibiotic complex. Endorsing the theory that RND-pump inhibitors act through an "altered-dynamics" mechanism, obstructing the translocation of substrates rather than preventing their binding and recognition.

Furthermore, we reveal that an MDR-conferring AcrB drug-pocket substitution, G288D[30], modifies the structural dynamics of the translocation pathway in a substrate-independent manner. Our previous MD simulations[30] show that the G288D substitution increases the gyration radius and hydration at and around the DBP. This disruption could subsequently lead to the observed allosteric action on farther AcrB regions. In turn, these changes may alter the energetic barrier for substrate binding and transport during functional rotation and, consequently, be the ultimate cause for the altered substrate efflux caused by this mutation.

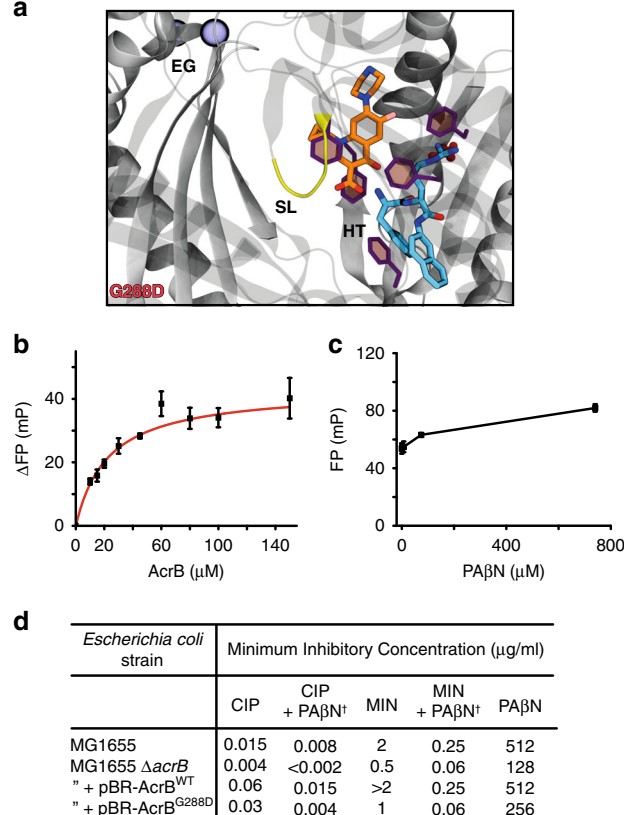

**Fig. 5 AcrB$^{G288D}$ is inhibited by the EPI PAβN. a** Molecular docking and multi-copy μs-long MD simulations reveal stable interactions of CIP (orange) and PAβN (cyan) to AcrB$^{G288D}$ T-state monomer and show their likely binding locations. The pose and its orientation are the same as shown for AcrB$^{WT}$ in Fig. 3c. EG = exit channel gate (blue spheres), SL = switch-loop (yellow), and HT = hydrophobic trap (purple). All computational data, including binding free energies can be found in Supplementary Table 5 and Supplementary Figs. 8, 14–16. **b** Binding of CIP by AcrB$^{G288D}$ in the presence of 150 μM of PAβN as determined by a fluorescence polarization assay performed by Su et al.[39]. All data are fit as in Fig. 3a ($R^2 = 0.99$). Data are the average and standard deviation from independent measurements ($n = 3$). **c** Binding competition assay between PAβN and CIP for AcrB$^{WT}$. Data are the average and standard deviation from independent measurements ($n = 3$). **d** MIC assays of *Escherichia coli* containing AcrB$^{WT}$ or AcrB$^{G288D}$ in the presence of PAβN and antibiotics. AcrB was overexpressed in MG1655 Δ*acrB* from a pBR322 plasmid containing its corresponding *acrAB* genes, natural promoter and "marbox" sequence. Minocycline = MIN, ciprofloxacin = CIP, and phenylalanine-arginine-β-naphthylamide = PAβN. †PAβN was added at a concentration of 50 μg/ml.

| *Escherichia coli* strain | Minimum Inhibitory Concentration (μg/ml) | | | | |
|---|---|---|---|---|---|
| | CIP | CIP + PAβN† | MIN | MIN + PAβN† | PAβN |
| MG1655 | 0.015 | 0.008 | 2 | 0.25 | 512 |
| MG1655 Δ*acrB* | 0.004 | <0.002 | 0.5 | 0.06 | 128 |
| " + pBR-AcrB$^{WT}$ | 0.06 | 0.015 | >2 | 0.25 | 512 |
| " + pBR-AcrB$^{G288D}$ | 0.03 | 0.004 | 1 | 0.06 | 256 |

conformation remaining the same even when the surrounding membranes display different curvatures[43]. This coupled with the agreement found between our HDX-MS data and MD simulations support that the structural dynamic behaviour uncovered here informs on the native protein state, although further studies investigating the effect of lipids on its dynamics are necessary to understand the system in its entirety.

We anticipate that the findings reported here will be important not only for establishing the general role of structural dynamics in modulating AcrB multidrug binding and efflux, which are hard to elucidate from biochemical and high-resolution structural data alone, but also for defining how naturally occurring mutations and EPI interactions affect its structure function.

## Methods

**Plasmid construction**. An overexpression plasmid containing AcrB with a C-terminal 6xHistidine tag (AcrB-6xHis) was constructed from a pET15b-AcrB-sGFP-6xHis plasmid from Reading et al.[44]. Briefly, the sGFP sequence was deleted and a 6xHis-tag placed at the C-terminus of AcrB followed by a stop codon using the Q5® site-directed mutagenesis kit (New England Biolabs)—AcrB contains two Histidine residues at its C-terminus, therefore, this construct resulted in AcrB having an 8xHistidine tag. The G288D mutation was then generated from this pET15b-AcrB-6xHis plasmid using the Q5® site-directed mutagenesis kit (New England Biolabs). All constructs were verified by DNA sequencing (Eurofins MWG). Primers used are reported in Supplementary Table 9.

pBR322-AcrB plasmids were generated for bacterial susceptibility assays. Briefly, pBR322 was linearized with HindIII and EcoRI restriction enzymes (New England Biolabs). *acrAB* genes with its natural promoter, including the "marbox" sequence, was then amplified from K-12 *Escherichia coli* chromosomal DNA (Zyagen Labs) and cloned into the pBR322 vector using In-Fusion® HD cloning (Takara Bio). A 6xHistidine tag sequence was included in the reverse primer to provide a 6xHis tag at the C-terminus of AcrB (pBR-AcrB$^{WT}$). The G288D mutation was generated from this pBR-AcrB$^{WT}$ plasmid using the Q5® site-directed mutagenesis kit (New England Biolabs). All constructs were verified by primer walking DNA sequencing (Eurofins MWG). Primers used are reported in Supplementary Table 9.

**AcrB overexpression and purification**. pET15b-AcrB-6xHis plasmid containing AcrB wildtype (AcrB$^{WT}$) or G288D mutant (AcrB$^{G288D}$) was transformed into C43 (DE3)Δ*acrB::KanR E. coli* cells. 7 ml of an overnight LB culture was added to 1 L of pre-warmed LB culture containing 100 μg/ml ampicillin and 30 μg/ml kanamycin and grown at 37 °C until an OD of 0.6–0.8 was reached. The culture was induced with 1 mM IPTG and grown for 16–18 h at 18 °C. At which point the cells were harvested by centrifugation at 4200 × *g* for 30 min and washed with ice-cold phosphate buffer saline (PBS).

Cell pellets were immediately resuspended in buffer A (50 mM sodium phosphate, pH 7.4, 300 mM sodium chloride) and supplemented with a protease inhibitor tablet (Roche), 100 μM PMSF, 1 μl Benzonase, and 5 mM beta-mercaptoethanol (β-ME). The cell suspension was then passed twice through a microfluidizer processor (Microfluidics) at 25,000 psi and 4 °C. Insoluble material was removed by centrifugation at 20,000 × *g* for 30 min at 4 °C. Membranes were then pelleted from the supernatant by centrifugation at 200,000 × *g* for 1 h at 4 °C. Membrane pellets were resuspended to 40 mg ml$^{-1}$ in ice-cold buffer A supplemented with a protease inhibitor tablet (Roche) and 100 μM PMSF, and homogenized using a Potter-Elvehjem Teflon pestle and glass tube.

AcrB was extracted from homogenized membranes by overnight incubation with 1% (w/v) *n*-dodecyl-β-D-maltoside (DDM) detergent (Anatrace) at 4 °C with gentle agitation. Insoluble material was then removed by centrifugation at 100,000 x *g* for 1 h at 4 °C. The sample was then filtered through a 0.22 μm filter (Fisher Scientific) and loaded onto a 1 ml HiTrap column (GE Healthcare) equilibrated in buffer B (50 mM sodium phosphate, pH 7.4, 300 mM sodium chloride, 20 mM imidazole, 10% (w/v) glycerol, 0.03% (w/v) DDM). The column was washed with 5 CVs of buffer B and then with 10 CVs of buffer C (50 mM sodium phosphate, pH 7.4, 300 mM sodium chloride, 20 mM imidazole, 10 % (w/v) glycerol, 1% (w/v) octyl glucose neopentyl glycol (OGNG) (Generon))—an OGNG detergent wash facilitates the removal of bound lipopolysaccharide (LPS), as determined previously[45] (Supplementary Fig. 2c-e). The column was then washed with 20 CVs of Buffer B containing 50 mM imidazole. AcrB was then eluted with Buffer B containing 500 mM imidazole and directly injected onto a Superdex 75 10/600 GL size exclusion chromatography (SEC) column (GE Healthcare) equilibrated in buffer D (50 mM sodium phosphate, pH 7.4, 150 mM sodium chloride, 10% (w/v) glycerol, 0.03% (w/v) DDM). A flow rate of 1 ml/min was used during HiTrap and SEC purification. Peak fractions eluted from the SEC column containing pure AcrB were pooled, spin concentrated using a 100 K MWCO concentrator (Amicon®), and spin filtered before being flash frozen and stored at −80 °C. SDS-PAGE electrophoresis was used to assess AcrB purification and protein concentration was

Here, to examine the structural dynamics of AcrB we performed HDX-MS within DDM detergent micelles and supported our findings with MD simulations completed within POPE/POPG (2:1 ratio) lipid bilayers. Detergent and amphipol membrane mimetics have been used extensively to obtain structural information for the determination of drug binding interactions and efflux mechanisms of AcrB. However, these systems do not provide a lipid environment, which can modulate membrane protein structure and function[40]. Recent studies have used SMA lipid particle (SMALP) technology and liposome reconstitution to capture AcrB within a lipid environment and solved its structure at high-resolution using cryo-EM[41–43]. The resulting structures were largely consistent with high-resolution crystal structures solved in DDM detergent micelles, with homotrimeric AcrB

calculated using a Cary 300 Bio UV–Vis spectrophotometer (Varian) with a calculated extinction coefficient[46] of $\varepsilon_{280} = 89{,}730\ M^{-1}\ cm^{-1}$.

**Circular dichroism spectroscopy.** Circular dichroism spectroscopy (CD) spectra were recorded with an Aviv Circular Dichroism spectrophotometer, Model 410 (Biomedical Inc., Lakewood, NJ, USA), with specially adapted sample detection to eliminate scattering artefacts. Multiple CD scans were averaged, the buffer background subtracted, and zeroed and minimally smoothed using CDTool[47]. A final protein concentration of 0.5–1.5 mg ml$^{-1}$ was used in a quartz rectangular or circular Suprasil demountable cell (Hellma Analytics). For thermal protein unfolding the mean residue ellipticity at 222 nm was monitored with increasing temperature.

**Native mass spectrometry.** Purified AcrB was buffer exchanged into MS buffer (200 mM ammonium acetate, pH 7.4, 0.03% (w/v) DDM or Triton X-100) using a centrifugal buffer exchange device (Micro Bio-Spin 6, Bio-Rad) according to the manufacturer's instructions[48]. Native mass spectrometry experiments were performed either on a Synapt G2-Si mass spectrometer (Waters) or a Thermo Scientific Q Exactive UHMR hybrid Quadrupole-orbitrap mass spectrometer.

For experiments on the Synapt G2-Si mass spectrometer the instrument settings used were: 1.5 kV capillary voltage, source temperature of 25 °C, argon trap collision gas, 180 V trap collision voltage, 120 V cone voltage, and 50 V source offset. Data were processed and analyzed using MassLynx v.4.1 (Waters).

Native mass spectrometry data on a Thermo Scientific Q Exactive UHMR hybrid Quadrupole-orbitrap mass spectrometer was acquired at resolving power 8750 at $m/z$ 400 in the $m/z$ range 2000–30.000. The instrument was optimized for transmission and desolvation of integral membrane proteins. Critical parameters throughout were relative pressure of 6, capillary temperature 250 °C, S-Lens RF level 0, in-source trapping 200 and the HCD energy was 300%. Data were analyzed by the use of Xcalibur software 4.3 and Biopharma Finder 3.1 (both Thermo Fisher Scientific). Deconvoluted spectra were acquired using Biopharma Finder 3.1 in sliding window mode using the following settings: Output mass range 10,000–100,000 Da, deconvolution mass tolerance 10 ppm, sliding window merge tolerance 30 ppm and minimal number of detected intervals.

**Preparation of ligands for hydrogen/deuterium mass spectrometry.** Ciprofloxacin (CIP) antibiotic and Phe-Arg-β-naphthylamide dihydrochloride (PAβN) inhibitor were both purchased from Sigma Aldrich. Stock concentrations of CIP (10 mg/ml) and PAβN (10 mg/ml) were prepared in 0.1 N HCl and water, respectively. As demonstrated previously[49], a primary consideration before carrying out HDX-MS is to ensure close to full binding to the target protein under deuterium exchange conditions; particularly for low-affinity ligands (dissociation constants, $k_D$, in the μM range or lower). CIP and PAβN both possess moderate affinity binding to AcrB within DDM detergent micelles (CIP with a $k_D$ of 74.1 ± 2.6 μM, as measured by fluorescence polarization[39] and PAβN with a $k_D$ of 15.72 ± 3.0 μM, as measured by surface plasmon resonance[50]. To achieve sufficient ligand binding saturation under deuterium exchange conditions AcrB was first incubated in the presence of saturating concentrations of ligand for 30 minutes on ice, before dilution into deuterated buffer containing saturating concentrations of ligand (740 μM, ~1000:1 ligand to AcrB ratio) for deuterium exchange experiments.

**Hydrogen/deuterium mass spectrometry.** Hydrogen/deuterium mass spectrometry (HDX-MS) was performed on an HDX nanoAcquity ultra-performance liquid chromatography (UPLC) Synapt G2-Si mass spectrometer system (Waters Corporation). Optimized peptide identification and peptide coverage for AcrB was performed from undeuterated controls. The optimal sample workflow for HDX-MS of AcrB was as follows: 5 μl of AcrB (15 μM) was diluted into 95 μl of either buffer D or deuterated buffer D at 20 °C. After fixed times of deuterium incubation samples were mixed with 100 μl of formic acid-DDM quench solution to provide a quenched sample at pH 2.5 and final 0.075% (w/v) DDM concentration. 80 μl of the quenched sample was then loaded onto a 50 μl sample loop before being injected onto an online Enzymate™ pepsin digestion column (Waters) in 0.1 % formic acid in water (at a flow rate of 200 μl/min) maintained at 20 °C. The peptic fragments were trapped onto an Acquity BEH C18 1.7 μM VANGUARD pre-column (Waters) for 3 min. The peptic fragments were then eluted using an 8–40% gradient of 0.1% formic acid in acetonitrile at 40 μl/min into a chilled Acquity UPLC BEH C18 1.7 μM 1.0 ×100 mm column (Waters). The trap and UPLC columns were both maintained at 0 °C. The eluted peptides were ionized by electrospray into the Synapt G2-Si mass spectrometer. MS$^E$ data was acquired with a 20–30 V trap collision energy ramp for high-energy acquisition of product ions. Argon was used as the trap collision gas at a flow rate of 2 mL/min. Leucine enkephalin was used for lock mass accuracy correction and the mass spectrometer was calibrated with sodium iodide. The online Enzymate™ pepsin column was washed three times with pepsin wash (1.5 M Gu-HCl, 4 % MeOH, 0.8% formic acid), as recommended by the manufacturer, and a blank run was performed between each sample to prevent significant peptide carry-over between runs.

All deuterium time points and controls were performed in triplicate. Sequence identification was performed from MS$^E$ data of digested undeuterated samples of AcrB using the ProteinLynx Global Server 2.5.1 software. The output peptides were

then filtered using DynamX (v. 3.0) using the following filtering parameters: minimum intensity of 1000, minimum and maximum peptide sequence length of 4 and 25, respectively, minimum MS/MS products of 1, minimum products per amino acid of 0.12, and a maximum MH$^+$ error threshold of 15 ppm. Additionally, all the spectra were visually examined and only those with a suitable signal to noise ratios were used for analysis. The amount of relative deuterium uptake for each peptide was determined using DynamX (v. 3.0) and was not corrected for back exchange[32]. The relative fractional uptake (RFU) was calculated from RFU$_a$ = [$Y_{a,t}$/ (MaxUptake$_a$ × D)], where Y is the deuterium uptake for peptide a at incubation time t, and D is the percentage of deuterium in the final labelling solution.

Confidence intervals for differential HDX-MS (ΔHDX) measurements of any individual time point were then determined according to Houde et al.[51] using Deuteros software (v. 1.0)[34]. There was no correlation found between ΔHDX values and their standard deviations ($R^2 = 0.09$). Only peptides which satisfied a ΔHDX confidence interval of 98% were considered significant. All ΔHDX AcrB structure figures were generated from the data using Deuteros in-house source code/ software[34] and Pymol[52]. All supporting data and meta-data are reported in the Source data file and Supplementary Data 1-4.

The mass spectrometry proteomics data have been deposited to the ProteomeXchange Consortium via the PRIDE partner repository with the dataset identifier PXD019047 [http://www.ebi.ac.uk/pride/archive/projects/PXD019047].

**Bacterial susceptibility assays.** Antimicrobial susceptibility testing of all wildtype and mutant strains in the absence or presence of PAβN was determined in triplicate by the broth microdilution (BMD) method and the standardized agar doubling-dilution method as recommended by the European Committee on Antimicrobial Susceptibility Testing (EUCAST). EUCAST guidelines were followed conforming to ISO 20776-1:2006[53,54]. Antibiotics and EPIs were made up and used according to the manufacturer's instructions. *E. coli* ATCC 25922 was used as the control strain.

**Fluorescence polarization.** AcrB ligand binding was determined using fluorescence polarization (FP) assays as performed by Su et al.[39]. An AcrB–PAβN protein complex stock was prepared; to ensure a loaded complex, AcrB and 150 μM PAβN was incubated for 2 h at 25 °C before titrating with 1.5 μM CIP ($k_D$ of PAβN is 15.72 ± 3.0 μM, as measured by surface plasmon resonance[50]). AcrB protein titration experiments were performed in ligand binding solution (50 mM sodium phosphate, 150 mM sodium chloride, 10% (v/v) glycerol, 1.5 μM CIP, 150 μM PAβN, 0.03% (w/v) DDM, pH 7.4). FP measurements were taken after incubation for 5 min for each corresponding protein concentration to ensure that the binding has reached equilibrium. Ligand binding data was fit to a hyperbola function (FP = ($B_{max}$*[protein])/($k_D$ + [protein])) as performed previously by Su et al.[39] using ORIGIN Ver. 7.5. (OriginLab Corporation, Northampton, MA, USA).

A FP competition assay was performed by titrating increasing concentrations of PAβN (0–1000 μM) to a AcrB–CIP preformed complex concentration adjudged from the binding data (1.5 μM CIP and 45 μM AcrB)—CIP and AcrB were maintained at the same concentration during all PAβN titrations. FP measurements were taken after incubation for 5 min for each corresponding protein concentration to ensure that the binding has reached equilibrium.

Each data point was an average of 15 FP measurements and each titrations series was performed three times. The absorption spectra of PAβN from 350 to 500 nm exhibited that PAβN absorbs light at 350 and 370 nm. The excitation wavelength of CIP at 415 nm does not excite PAβN, therefore PAβN can be considered as a non-fluorescent ligand within these experiments. DDM detergent concentration was consistent to eliminate possible changes in polarization by drug–DDM micelle interactions.

**Molecular docking.** A blind docking campaign was first performed using Autodock Vina[55]. As done in Atzori et al.[56], a rectangular search space of size 125 Å × 125 Å × 110 Å enclosing the whole portion of the protein potentially exposed to ligands was adopted. The exhaustiveness parameter, related to the extent of the exploration within the search space, was set to 8192 (~1000 times the default 8) in order to improve the sampling of docking poses within the large box used (~64 times the default 30 Å × 30 Å × 30 Å). Flexibility of both partners was considered indirectly, by employing multiple conformations in ensemble docking runs[57]. For both CIP and PAβN, 10 representative molecular conformations were obtained from 1 μs long molecular dynamics simulations of the compounds in the presence of explicit solvent[58] (data available at www.dsf.unica.it/translocation/db). Namely, a cluster analysis of the trajectories of the ligands was performed as described in Malloci et al.[58], setting the number of cluster representatives to 10.

For the wildtype receptor (AcrB$^{WT}$), 10 X-ray asymmetric high-resolution structures (with PDB IDs: 2GIF, 2DHH, 2J8S, 3W9I, 4DX5, 4DX7, 4U8V, 4U8Y, 4U95, 4U96) were considered, most bearing a substrate bound to the transporter. For the G288D variant of AcrB (AcrB$^{G288D}$), we also employed 10 structures, namely the homology models derived on top of the AcrB$^{WT}$ X-ray structures mentioned above. Regarding the homology modelling protocol, the sequence of the G288D variant was first generated by manually modifying the FASTA file of the corresponding amino acid sequence of *E. coli* AcrB retrieved from the Uniprot database (Uniprot Id: P31224). Next, 100 homology models were generated for

each template with the Modeller 9.21[59] software. The variable target function method was used to perform the optimization, and the best model (that is the one with the highest value of the MOLPDF function) was employed in docking calculations.

The ensemble docking campaign resulted in several hundred poses per ligand, most of which were located inside the DBP of the monomer in the T-state (DBP$_T$), which is the putative binding site for the recognition of low molecular mass compounds such as those studied here[11] (see Supplementary Table 1). Because most docking poses were concentrated in this region, we performed a second docking campaign using a grid of 30 Å × 30 Å × 30 Å and centred at DBP$_T$. Next, we performed a cluster analysis of the docking poses using as a metric the heavy atoms Root Mean Square Deviation (RMSD) of the substrate (setting the cut-off to 3 Å), which returned respectively 11, 9, 15, and 17 different poses for the AcrB$^{WT}$–PAβN, AcrB$^{WT}$–CIP–PAβN, AcrB$^{G288D}$–PAβN, AcrB$^{G288D}$–CIP–PAβN complexes (Supplementary Tables 2, 5). Moreover, to evaluate how the presence of PAβN affects the binding of CIP in the ternary complexes, we selected three docking poses of CIP onto AcrB$^{WT}$ and AcrB$^{G288D}$. In the case of the AcrB$^{WT}$–CIP complex, to consider the largest number of putative binding modes, we purposely selected docking poses with an orientation different than that reported previously[29] (Supplementary Fig. 16).

**Molecular dynamics simulations.** All of the 52 complexes selected from docking runs were subjected to all-atom molecular dynamics (MD) simulations (each of 1 μs in length) performed with the AMBER18 package[60].

Protomer-specific protonation states of AcrB were adopted following previous work[61]: residues E346 and D924 were protonated only in the L and T protomers, while residues D407, D408, and D566 were protonated only in the O protomer, of AcrB. The topology and the initial coordinate files were created using the LEaP module of the AMBER18 package. The proteins were embedded in a mixed bilayer patch composed of 1-palmitoyl-2-oleoyl-*sn*-glycero-3-phosphoethanolamine (POPE) and 1-palmitoyl-2-oleoyl-*sn*-glycero-3-phosphoglycerol (POPG) in a 2/1 ratio, for a total of 660 lipid molecules symmetrically distributed in the two leaflets of the bilayer. The whole system was solvated with a 0.15 M aqueous NaCl solution. The AMBER force-field protein.fb15[62] was used to represent the protein; lipid17 (http://ambermd.org/GetAmber.php) parameters were used for the POPE molecules; the TIP3PFB model was employed for water[63]. The General Amber Force-Field (GAFF) parameters[64] for CIP and PAβN were taken from Malloci et al[58].

Each system was first subjected to a multi-step structural relaxation via a combination of steepest descent and conjugate gradient methods using the *pmemd* program implemented in AMBER18, as described in previous publications[4,29,61]. The systems were then heated from 0 to 310 K in two subsequent MD simulations: (i) from 0 to 100 K in 1 ns under constant-volume conditions and with harmonic restraints ($k = 1$ kcal mol$^{-1}$·Å$^{-2}$) on the heavy atoms of both the protein and the lipids; (ii) from 100 to 310 K in 5 ns under constant pressure (set to a value of 1 atm) and with restraints on the heavy atoms of the protein and on the z coordinates of the phosphorous atoms of the lipids to allow membrane rearrangement during heating. As a final equilibration step, a series of 20 equilibration steps, each of which was 500 ps in duration (total 10 ns), with restraints on the protein coordinates, were performed to equilibrate the box dimensions. These equilibration steps were carried out under isotropic pressure scaling using the Berendsen barostat, whereas a Langevin thermostat (collision frequency of 1 ps$^{-1}$) was used to maintain a constant temperature. Finally, production MD simulations of 1 μs were performed under an isothermal-isobaric ensemble for each system. A time step of 2 fs was used for all runs before production, while the latter runs were carried out with a time step of 4 fs after hydrogen mass repartitioning[65].

During the MD simulations, the lengths of all the R–H bonds were constrained with the SHAKE algorithm. Coordinates were saved every 100 ps. The Particle mesh Ewald algorithm was used to evaluate long-range electrostatic forces with a non-bonded cut-off of 9 Å.

**Post-processing of MD trajectories.** MD trajectories were analyzed using either in-house *tcl* and *bash* scripts or the *cpptraj* tool of AMBER18. Figures were pre-pared using gnuplot 5.0[66] and VMD 1.9.3[67]. All the calculations with the exception of the cluster analysis were performed on the conformations taken from the most populated conformational cluster (representing the most sampled conformation of the complex along the production trajectories) along the last 300 ns of the production runs.

*Cluster analysis.* Clustering of the ligand trajectory was carried out using the average-linkage hierarchical agglomerative clustering method implemented in *cpptraj* and employing an RMSD cut-off of 3 Å calculated on all the heavy atoms of the ligand.

*System stability.* The RMSDs of the protein and of the substrates were calculated using *cpptraj* after structural alignment of each trajectory (reported in Supplementary Table 10). Namely, we calculated the Cα-RMSD of the protein with respect to the initial (docking) structure after alignment of the whole trimer. The RMSDs of the substrates were calculated with respect to the corresponding structure of the selected docking pose, as well as with respect to the last frame of the MD trajectory. In particular. To evaluate the magnitude of the displacements and reorientations of the substrates during the simulations, their RMSDs were calculated upon alignment of the T monomer of the protein to the reference frame.

*Interaction network.* Interactions stabilizing the complexes were analyzed by considering residues within 3.5 Å of each substrate in the last 300 ns of the MD trajectories. Hydrogen bonds were identified through geometrical criteria, using a cut-off of 3.2 Å for the distance between donor and acceptor atoms and a cut-off of 135° for the donor-hydrogen-acceptor angle. Such analyses were conducted through *in-house* tcl scripts. Occupancy levels of hydrogen bonds and water-mediated interactions (detected in the last 300 ns of each simulation) were also computed using *cpptraj*. For systems AcrB$^{WT}$–PAβN, AcrB$^{WT}$–CIP-PAβN, AcrB$^{G288D}$–PAβN and AcrB$^{G288D}$–CIP-PAβN, the following analyses were also performed to evaluate their agreement with HDX-MS data.

*System flexibility.* The Root Mean Square Fluctuations (RMSFs) of the protein were calculated using *cpptraj* after structural alignment of each trajectory as described in the previous paragraph.

*Hydration properties.* Residue-wise average numbers of waters within the first (second) hydration layer were calculated with *cpptraj* using a distance cut-off of 3.4 (5.0) Å between the nitrogen of the protein and the water oxygens.

*Comparison with HDX-MS data.* RMSFs and hydration properties of each system were compared with a proper reference state according to the current knowledge about the most likely conformations assumed by AcrB in the absence of ligands or complexed with substrates and inhibitors[16]. For instance, to account for conformational changes of AcrB induced by inhibitor binding, PAβN-bound and apo AcrB structures were considered in their T- and L-state, respectively. The T-state was also considered for systems containing both PAβN and CIP (AcrB$^{WT}$–CIP–PAβN and AcrB$^{G288D}$–CIP–PAβN), hypothesizing their stability in this conformation, as evidenced by the RMSDs analyses conducted on our trajectories (Supplementary Figs. 6, 9, 12, 14). The list of reference states used for each analysis are reported in Supplementary Table 11.

**Reporting summary.** Further information on research design is available in the Nature Research Reporting Summary linked to this article.

## Data availability

Data supporting the findings of this manuscript are available from the corresponding authors upon reasonable request. The molecular dynamics trajectories will be available anyone at any time by sending an e-mail to Attilio Vittorio Vargiu (vargiu@dsf.unica.it). A reporting summary for this Article is available as a Supplementary Information file. Mass spectrometry data files including processed DynamX files have been deposited to the ProteomeXchange Consortium via the PRIDE partner repository with the dataset identifier PXD019047. HDX-MS meta-data have been provided in Supplementary Data files 1–4 with this paper. Source data are provided with this paper.

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

## Acknowledgements

Work at King's College London was supported by a UKRI Future Leaders Fellowship (MR/S015426/1) to E.R., the Wellcome Trust (109854/Z/15/Z) to A.P., a Wellcome Trust Investigator Award 214259 to P.J.B., and a King's Health Partners R&D Challenge Fund through the MRC Confidence in Concept Grant (MC_PC_15031) to A.P., E.R., and P.J.B. Work at the University of Cagliari received support from the National Institutes of Allergy and Infectious Diseases project number AI136799 (C.F., G.M., P.R., and A.V.V.) and the Innovative Medicines Initiatives Joint Undertaking under grant agreement number 115525 (C.F., G.M., P.R., and A.V.V.), resources that are composed of financial contribution from the European Union 7th Framework Programme (FP7/2007-2013) and EFPIA companies in kind contribution. Work at the University of Birmingham by L.J.V.P., X.W.K., J.S., and V.R. was supported by MRC grant, (MR/P022596/1) and E.M.G. by an MRC iCASE studentship (MR/N017846/1). We thank Bram Snijders, Peter Faull and Shahid Mehmood (Francis Crick Institute, London) for the use of their Thermo Scientific Q Exactive UHMR hybrid Quadrupole-orbitrap mass spectrometer, and Andrea Bosin and Giovanni Serra (University of Cagliari) for technical assistance with the use of HPC resources. We thank H. Zgurskaya and O. Lomovskaya for enlightening discussions during the revision of the manuscript.

## Author contributions

E.R., Z.A., A.V.V, L.J.V.P., and A.P. designed the research. E.R., Z.A., A.M.L., and H.F. performed all experiments and analyses, except for molecular modelling and bacterial susceptibility assays. C.F., G.M., and A.V.V. carried out docking, molecular dynamics and performed post-molecular dynamics analyses. X.W.K., V.R., J.S., and E.M.G. performed bacterial susceptibility assays and analysis. A.K. and Z.A. performed UHMR Native MS measurements and analysis. E.R., P.J.B., P.R., A.V.V., L.J.V.P., and A.P. supervised the project. E.R., Z.A., and A.P. wrote the manuscript with input from the other authors.

## Competing interests

The authors declare no competing interests.
