## [Peer Review File · Nature Communications]

REVIEWER COMMENTS

Reviewer #1 (Remarks to the Author):

In this manuscript the authors describe their efforts to determine the structural dynamics of the drug efflux pump AcrB in the presence of substrates and the efflux pump inhibitor PABN. Complimentary methods of hydrogen/deuterium exchange mass spectrometry and molecular modelling were used to determine the structural dynamics of inhibition and transport. This is a novel approach and yielded interesting data. The structural dynamics were also investigated for AcrB from a MDR clinical isolate with a G288D substitution in the drug binding pocket. This is an accomplished work that would make an important contribution to the field.

The hydrogen/deuterium exchange mass spectrometry were determined for AcrB purified and solubilised in the detergent DDM. Detergent solubilised protein is not necessarily the best representation of the native protein state. The authors should discuss if they would expect differences if the protein were solubilised with SMA technology or reconstituted in a lipid membrane.

The result with PABN and PABN-Ciprofloxacin combination is very interesting. PABN does not have a particularly high binding affinity for AcrB in comparison with other inhibitors or even substrates though. Would the result be different if another, more high affinity, inhibitor was used instead of PABN?

It is interesting that PABN completely abolish ethidium bromide resistance in AcrB from Escherichia coli. However, PABN does not compete with ethidium bromide in transport assays in Pseudomonas aeruginosa. What would be the explanation for this difference?

Reviewer #2 (Remarks to the Author):

Reading et al. examine closely the structural dynamics of AcrB by means of molecular dynamics simulations and H/D exchange mass spectrometry. The bottom line conclusion from their studies is that the PABN efflux pump inhibitor restricts the intrinsic motions of the drug-binding pockets, indicating that structural dynamics play a critical role in the inhibition and substrate specificity of AcrB. The idea is that PABN does not prevent antibiotic binding but rather inhibits efflux by enforcing a restrained state that reduces the frequency and magnitude of the conformational changes in the substrate translocation path. The impressive amount of work reported appears to be competently done and accurately reported.

The work presents a new point of view that specialists will be interested in. I doubt that more casual readers will be willing to slug through the paper to understand the findings. The paper is complex and relies heavily upon supplementary materials. For example, reading carefully the 'supplementary discussion' is mandatory for appreciating the paper. The paper is more appropriate for a specialist journal.

A minor but irritating point is the overuse of the word 'impact', which the authors use both as a verb and a noun. This has become rampant in the literature. The better usage is affect (verb) and effect (noun).

Reviewer #3 (Remarks to the Author):

In this work, the authors use HDX-MS and MD simulations to investigate the synergistic effect of substrate (CIP) and inhibitor (PABN) binding to the MDR efflux pump AcrB, together with a clinically

relevant mutation (G288D).

The three main findings are clearly stated in the introduction:

- PA β N EPI restricts the intrinsic motions of the drug-binding pockets as part of its mechanism of action and is effective against both AcrBWT and AcrBG288D
- an EPI can dually bind to AcrB alongside an antibiotic, without affecting its inhibitory action
- an MDR mutation in *acrB* impacts upon the structural dynamics of the efflux translocation pathway likely contributing to its modified substrate specificity

The authors used adequate methods (i.e. CD and thermal protein unfolding) to confirm that both recombinant constructs (WT and G288D) have similar overall structural and thermal stability. The article is particularly well written, adapted to the broad readership of *Nature Communications* and yet giving enough details in the supplementary material for structural biologists (concerning the analysis of the MD simulations in particular).

I believe that this article should be considered for publication in *Nature Communications* with editorial revisions.

Suggestions below are designed to enhance its communication and impact for the non-specialist.

Major comments:

- Supplementary Fig. 2c-d: Different MWs were obtained native MS before and after delipidation. Could the authors comment on that?
- l.138: "antibiotic binding only subtly alters the structural motions of AcrBWT": how can the authors make sure that they have a stoichiometric binding and that the binding is not just transient...this sentence is possibly overstated or at least too general as the results might be different at different concentrations and for different antibiotics... The reviewer suggests to replace "antibiotics" by "CIP".
- The sequence coverage seems different in the different conditions (CIP or PA β N), could the authors comment on that? On the same line, did the mutation induce a change in the sequence coverage of the region carrying the G288D mutation? Were the authors able to compare the peptide(s) carrying the mutation?
- Simulations show the formation of hydrophobic bonds between PA β N and E130/K131/Q176/A619 (pose 1), E130/D276 (pose 2) and S126/S46/D174 (pose 3). How do the HDX-MS protection upon PA β N binding relate to these different proposed H-bonds? Are there any encompassing peptide significantly protected by PA β N? If yes this should be clearly stated in the text. If not, do the authors think that this might be due to a partial and/or transient binding of the drug? This point should be addressed in the manuscript.
- Fig. 2 shows the Δ HDX comparisons of (AcrBWT+CIP) vs. (AcrBWT) and (AcrBWT+PA β N) vs. (AcrBWT). We can clearly see that CIP mainly destabilizes the N-ter & C-ter of AcrBWT and stabilizes/protects the connecting-loop, while binding of PA β N additionally protects other regions, including the Switch loop. However, a proper statistical comparison between (AcrBWT+CIP) vs. (AcrBWT+PA β N) would be needed before stating that "However, in stark contrast to CIP, inhibitor binding led to HDX reduction throughout extensive parts of the PC1/PC2 cleft of the drug-binding pockets and within the connecting-loop (Fig. 2a-b). This could signify inhibitor induced structural stabilization of the drug-binding pocket entrances" (l.139-140).
- Fluorescence polarization binding and competition assays indicate that both molecules can interact independently with AcrB. This was further supported by MD simulations of AcrB in the Tstate with both CIP and PA β N nicely showing that PA β N is able to bind to the HT region while CIP lies next to the DBP-PBP interface. I just wonder how the CIP docking is affected by the additional presence of PA β N? Is CIP binding to the same site in the absence of PA β N? Could it be that the presence of PA β N stabilizes

one particular bound state of CIP, preventing its efflux? This seems to be suggested by the MD simulation analysis (supplementary material) but maybe not enough clearly stated in the main text.

- The authors then present in Fig.2 the Δ HDX comparison of (AcrBWT+CIP+PA β N) vs. (AcrBWT), which indeed shows very similar behavior as the previous comparison (AcrBWT+PA β N) vs. (AcrBWT), suggesting that most fluctuations are due to PA β N and that there isn't much synergy between the substrate and the EPI. However, to see the additional effect of each molecule, identify which regions are statistically altered synergistically and reach these conclusions, proper statistical comparisons between the states (AcrBWT+CIP+PA β N) vs. (AcrBWT+CIP) and (AcrBWT+CIP+ PA β N) vs. (AcrBWT+PA β N) are required (at least in the supplementary material).

- The local and allosteric effects of the G288D mutation on the dynamics of AcrB are quite impressive, as shown by the changes in HDX observed on Fig.4b (no drugs). Irrespective of the drug(s), the G288D mutation induces local flexibility (increase HDX) and a long-range stabilization (decrease HDX) of the connecting loop, as seen in Fig.4c. The effects of the drugs are first represented by comparing the HDX of AcrBG288D+CIP or PA β N with AcrBWT, meaning that two parameters (mutation and presence of the drug) are changing and that the contribution of each parameter is hard to apprehend. Actually, it seems to me that the local destabilization of the PN2 region is only due to the G288D mutation rather than by the drugs as it seems to be the case in Fig.4c. The authors then rightfully compare (AcrBG288D+CIP) vs. (AcrBG288D) and (AcrBG288D+PA β N) vs. (AcrBG288D) (supplementary Fig.9), which is better, I think to see the sole effect of each drug on the mutant. In this representation, the destabilization of the PN2 region by the drugs is not visible anymore. I would thus recommend to present these comparisons (AcrBG288D+drug) vs. (AcrBG288D) in the main text and the one from Fig.4c in the supplementary data (just keeping the AcrBG288D vs. AcrBWT comparison from Fig.4c), in order to avoid any misleading interpretation of the data. Alternatively, the authors could represent the comparison of (AcrBG288D+drug) vs. (AcrBWT+drugs) to see the effect of the mutation on the drug-bound AcrB (as mentioned later on lines 295-302).

- The comparison throughout the manuscript of the first hydration shell and HDX levels is very interesting. The authors claim several times that these results of simulations are in good agreement with their HDX data. I think it would be a good idea, for a non-expert readership, to detail a little bit more in what sense these results agree for eg. by saying (in the main text) that a reduced hydration shell should imply reduced HDX and why.

- On the same line, the fact that hydrogen bonds were found in the MD ternary complex simulations, between the two drugs (and not only with the HT and switch-loop) is, I think, important to state in the main text.

- The antibiotics used for MIC assays are different in Fig.2c and Fig.5c (apart from CIP), could the authors explain why?

- The last sentence of the results states that "G288D has a profound impact on AcrB substrate specificity within both E. coli and Salmonella", however the MD simulations clearly showed that both WT and G288D AcrB were able to bind to both CIP (with and w/o PA β N). I don't clearly see how the data presented here infers directly a difference in substrate "specificity" per se (in terms of Kd for eg.) but rather on substrate export (indirectly with the MIC assays) or stability (HDX and MD simulations). Could the authors use fluorescence polarization to determine and compare the Kds of CIP and PA β N to AcrBG288D as they did for the WT construct?

- The authors use a \sim 1000:1 ligand to AcrB ratio for the HDX experiments: how to make sure that in these conditions, non-specific binding does not occur? This could obviously affect the regions pinpointed by the HDX experiments. Do the authors expect to see a single drug bound to AcrB under such conditions in native MS?

- The HDX-MS data was made available through on Pride (#416950) but I could not find it. The authors need to make sure that the data is either publicly available or provide login and password to the reviewers. The deposited data should include .raw files, DynamX .csv output files, .fasta files and PLGS output files.

Minor Comments:

- Supplementary Fig.2e: please map the TM domains to the sequence.
- Supplementary Fig.3 shows the HDX heatmap obtained for AcrBWT alone. The reviewer suggests to include (panel C) a 3D structure color-coded with this HDX-MS data (last time point for eg.).
- l.283: "how the mutation effects" should read "how the mutation affects".
- The top legend of Supplementary Fig.9 should read (AcrBG288D+drug(s)) instead of AcrBG288D+/- drug(s)).

In the Material and Methods section:

- the flow rates for the protein purification (elution on HisTrap and SEC) should be indicated.
- the MWCO of the spin column for the concentration step should be indicated.
- the argon trap collision gas flow rate should be indicated.
- l.670: "Bipharma" should read "Biopharma".
- HDX: the volume of sample transferred to the quench solution, the volume of quench solution and the volume of sample injected (as well as the volume of the loop) should be indicated.

1st September 2020

Reviewer #1 (Remarks to the Author):

In this manuscript the authors describe their efforts to determine the structural dynamics of the drug efflux pump AcrB in the presence of substrates and the efflux pump inhibitor PA β N.

Complimentary methods of hydrogen/deuterium exchange mass spectrometry and molecular modelling were used to determine the structural dynamics of inhibition and transport. This is a novel approach and yielded interesting data. The structural dynamics were also investigated for AcrB from a MDR clinical isolate with a G288D substitution in the drug binding pocket. This is an accomplished work that would make an important contribution to the field.

The hydrogen/deuterium exchange mass spectrometry were determined for AcrB purified and solubilised in the detergent DDM. Detergent solubilised protein is not necessarily the best representation of the native protein state. The authors should discuss if they would expect differences if the protein were solubilised with SMA technology or reconstituted in a lipid membrane.

The reviewer makes a very good point as lipids are likely to play important roles in membrane protein structure and function. Studying AcrB within lipid systems, such as SMALPs, is likely to offer increased insight into its native structure/function. Though our HDX-MS investigations were performed within detergent micelles, our MD simulations were completed within a POPE/POPG (2:1 ratio) lipid bilayer. The agreement found between our HDX-MS data and MD simulations supports that the structural dynamic behaviour uncovered here informs on the native protein state. Nevertheless, experiments performed within a lipid environment to fully understand AcrB structure-function is the natural next step, and it will form part of our ongoing and future studies. An additional discussion regarding the above points and recent literature on cryo-EM of AcrB within SMALPs/liposomes has been added to the Discussion section:

“Here, to examine the structural dynamics of AcrB we performed HDX-MS within DDM detergent micelles and supported our findings with MD simulations completed within POPE/POPG (2:1 ratio) lipid bilayers. Detergent and amphipol membrane mimetics have been used extensively to obtain structural information for the determination of drug binding interactions and efflux mechanisms of AcrB. However, these systems do not provide a lipid environment, which can modulate membrane protein structure and function⁴⁰. Recent studies have used SMA lipid particle (SMALP) technology and liposome reconstitution to capture AcrB within a lipid environment and solved its structure at high-resolution using cryo-EM⁴¹⁻⁴³. The resulting structures were largely consistent with high

resolution crystal structures solved in DDM detergent micelles, with homotrimeric AcrB conformation remaining the same even when the surrounding membranes display different curvatures⁴³. This coupled with the agreement found between our HDX-MS data and MD simulations supports that the structural dynamic behaviour uncovered here informs on the native protein state, although further studies investigating the effect of lipids on its dynamics are necessary to understand the system in its entirety.”

New references added:

- 40 *Laganowsky, A. et al. Membrane proteins bind lipids selectively to modulate their structure and function. Nature 510, 172-175, doi:10.1038/nature13419 (2014).*
- 41 *Qiu, W. et al. Structure and activity of lipid bilayer within a membrane-protein transporter. Proceedings of the National Academy of Sciences 115, 12985, doi:10.1073/pnas.1812526115 (2018).*
- 42 *Johnson, R. M. et al. Cryo-EM Structure and Molecular Dynamics Analysis of the Fluoroquinolone Resistant Mutant of the AcrB Transporter from Salmonella. Microorganisms 8, 943, doi:10.3390/microorganisms8060943 (2020).*
- 43 *Yao, X., Fan, X., & Yan, N. Cryo-EM analysis of a membrane protein embedded in the liposome. Proc. Natl. Acad. Sci. U.S.A. 117, 18497-18503, doi: 10.1073/pnas.2009385117 (2020).*

The result with PA β N and PA β N-Ciprofloxacin combination is very interesting. PA β N does not have a particularly high binding affinity for AcrB in comparison with other inhibitors or even substrates though. Would the result be different if another, more high affinity, inhibitor was used instead of PA β N?

We posit that observed effects on AcrB dynamics by the PA β N inhibitor are general for inhibitors which bind to the hydrophobic trap but are likely to be more pronounced for a higher affinity inhibitor, due to their ability to lock the pump in a more homogeneous conformation (as was observed for the higher affinity inhibitor, MBX3132 (developed by Microbiotix Inc.), in: Wang, Z. et al. An allosteric transport mechanism for the AcrAB-TolC multidrug efflux pump. eLife 6, e24905, doi:10.7554/eLife.24905 (2017) – reference 16 in this manuscript). However, this is an extremely interesting point, which forms part of future work but is outside the scope of this study.

It is interesting that PA β N completely abolish ethidium bromide resistance in AcrB from Escherichia coli. However, PA β N does not compete with ethidium bromide in transport assays in Pseudomonas aeruginosa. What would be the explanation for this difference?

PA β N indeed does not potentiate the activity of ethidium bromide (EtBr) in a wildtype Pseudomonas strain (Lomovskaya, O et al. “Identification and characterization of inhibitors of multidrug resistance efflux pumps in Pseudomonas aeruginosa: novel agents

*for combination therapy.” Antimicrobial agents and chemotherapy vol. 45,1 (2001): 105-16. <https://www.ncbi.nlm.nih.gov/pmc/articles/PMC90247/>). The difference could be due to the markedly different permeabilities of the outer membranes of *E. coli* and *P. aeruginosa*, the latter being much more difficult to bypass (indeed, hyperporination does not affect the MICs of EtBr in *E. coli* but strongly affects the MICs in *P. aeruginosa* - H. Zgurskaya, private communication).*

*However, prompted by the point made by the reviewer, we further explored the literature and found that, while the data regarding the PA β N-induced MIC changes for CIP, TET and CHL are consistent with previous reports [see e.g. Table 2 in Machado et al. (2017), PeerJ 5:e3168; DOI 10.7717/peerj.3168; Fig. 1 in Schuster et al., Molecules 2019, 24, 470 (2019); Table 1 in Piddock et al., J Antimicrob Chemother 2010; 65: 1215–1223], those for EtBr are contrasting [see e.g. Table 2 in Machado et al. (2017), PeerJ 5:e3168; DOI 10.7717/peerj.3168, reporting a one-fold MIC change; whereas, EtBr accumulation assays performed by Fanning and co-workers – Fig. 1 in Karczmarczyk et al, APPLIED AND ENVIRONMENTAL MICROBIOLOGY, 2011, 7113 - reported increased intracellular concentration in the presence of PA β N]. Since the data regarding the potentiation of EtBr by AcrB^{WT} in *E. coli* are irrelevant for the main findings presented in this work, and considering that the COVID-19 pandemic limits us to explore this aspect further in the next months, we opted for removing them from Fig. 2 and the manuscript text. We plan to investigate in more detail our findings once we have full access to the laboratory.*

Reviewer #2 (Remarks to the Author):

Reading et al. examine closely the structural dynamics of AcrB by means of molecular dynamics simulations and H/D exchange mass spectrometry. The bottom line conclusion from their studies is that the PA β N efflux pump inhibitor restricts the intrinsic motions of the drug-binding pockets, indicating that structural dynamics play a critical role in the inhibition and substrate specificity of AcrB. The idea is that PA β N does not prevent antibiotic binding but rather inhibits efflux by enforcing a restrained state that reduces the frequency and magnitude of the conformational changes in the substrate translocation path. The impressive amount of work reported appears to be competently done and accurately reported.

The work presents a new point of view that specialists will be interested in. I doubt that more casual readers will be willing to slug through the paper to understand the findings. The paper is complex and relies heavily upon supplementary materials. For example, reading carefully the 'supplementary discussion' is mandatory for appreciating the paper. The paper is more appropriate for a specialist journal.

We used HDX-MS to study and assess the structural dynamics of AcrB in the presence of substrates for the first time. MD and docking studies were performed to provide evidence for a ternary protein-EPI-antibiotic complex and to provide further molecular-level insight

into substrate interactions within the drug-binding pockets of AcrB. We expanded our investigations to study the clinically relevant drug-pocket substitution, G288D, and showed that this mutation perturbs the structural dynamics along AcrB's substrate translocation pathway. Together with supporting biochemical and bacterial susceptibility assays, our findings reveal the directive role of protein structural dynamics in the structure-function of AcrB (a prototypical member of the RND family, which plays a key role in MDR in bacteria) in relation to inhibition and naturally occurring mutation. We appreciate the reviewer's comments that in our previous manuscript these broad findings were not expressed clearly. We have, therefore, re-focussed our discussion section of our manuscript to better explain the broader aspects of our conclusions – the more specific conclusions being stated in the end of the introduction (as mentioned by reviewer #3). The two re-worked paragraphs below have been included at the beginning and end of the Discussion section of the manuscript:

(Beginning) "In summary, we found that binding of an EPI, PAβN, restricts AcrB dynamics and could not be outcompeted by an antibiotic, CIP, whose activity it potentiates. Fluorescence binding, MD simulations and docking studies supporting the existence of a ternary protein-EPI-antibiotic complex. Endorsing the theory that RND-pump inhibitors act through an "altered-dynamics" mechanism, obstructing the translocation of substrates rather than preventing their binding and recognition.

Furthermore, we reveal that an MDR-conferring AcrB drug-pocket substitution, G288D³⁰, modifies the structural dynamics of its substrate translocation pathway in a substrate-independent manner. Our previous MD simulations³⁰ show that the G288D substitution increases the gyration radius and hydration at and around the DBP. This disruption could subsequently lead to the observed allosteric action on farther AcrB regions. In turn, these changes may alter the energetic barrier for substrate binding and transport during functional rotation and, consequently, be the ultimate cause for the altered substrate efflux caused by this mutation."

(End) "We anticipate that the findings reported here will be important not only for establishing the general role of structural dynamics in modulating AcrB multidrug binding and efflux, which are hard to elucidate from biochemical and high-resolution structural data alone, but also for defining how naturally occurring mutations and EPI interactions affect its structure-function."

We also agree that there is significant detail for the MD simulations and post-MD analysis within the Supporting Information. The focus of the manuscript was investigating the role of protein structural dynamics in the structure-function of AcrB, using HDX-MS and biochemical/bacterial assays. The MD investigations provide vital support to these conclusions, but we found that a dedicated supplementary discussion/material was

required to provide the rigorous assessment required for structural biologists to understand the specific substrate bonding interactions and the accordance of the MD simulations with the HDX-MS data. Having a dedicated discussion in the Supporting Information allows us to bring out our key findings in the main article text and best show the power of HDX-MS for understanding membrane protein structure-function to the broad readership of Nature Communications. However, we very much value the reviewer's comments and have attempted to better link the main text with the Supplementary Information throughout the manuscript in the submitted re-draft. We have also provided a Table of Contents for the Supplementary Information to better navigate the material.

Other material presented in the Supporting Information include essential protein characterization and additional differential HDX (Δ HDX) analysis (as requested by Reviewer #3 and/or requested by the terms set out in the recent HDX-MS 'white paper': Masson, G. R. et al. Recommendations for performing, interpreting and reporting hydrogen deuterium exchange mass spectrometry (HDX-MS) experiments. Nat. Methods 16, 595-602, doi:10.1038/s41592-019-0459-y (2019)).

A minor but irritating point is the overuse of the word 'impact', which the authors use both as a verb and a noun. This has become rampant in the literature. The better usage is affect (verb) and effect (noun).

The overuse of the word impact has been corrected within the text by replacement with affect/effect and/or alteration to sentence structure.

Reviewer #3 (Remarks to the Author):

In this work, the authors use HDX-MS and MD simulations to investigate the synergistic effect of substrate (CIP) and inhibitor (PA β N) binding to the MDR efflux pump AcrB, together with a clinically relevant mutation (G288D).

The three main findings are clearly stated in the introduction:

- PA β N EPI restricts the intrinsic motions of the drug-binding pockets as part of its mechanism of action and is effective against both AcrBWT and AcrBG288D
- an EPI can dually bind to AcrB alongside an antibiotic, without affecting its inhibitory action
- an MDR mutation in acrB impacts upon the structural dynamics of the efflux translocation pathway likely contributing to its modified substrate specificity

The authors used adequate methods (i.e CD and thermal protein unfolding) to confirm that both recombinant constructs (WT and G288D) have similar overall structural and thermal stability. The article is particularly well written, adapted to the broad readership of Nature

Communication and yet giving enough details in the supplementary material for structural biologists (concerning the analysis of the MD simulations in particular).

I believe that this article should be considered for publication in Nature communications with editorial revisions.

Suggestions below are designed to enhance its communication and impact for the non-specialist.

Major comments:

- Supplementary Fig.2c-d: Different MWs were obtained native MS before and after delipidation. Could the authors comment on that?

Multiple trimeric AcrB-LPS bound species present in the 'lipidated' spectra, identified by additional masses of 3.2–3.7 kDa from the apo form, resulted in a high degree of spectral overlap between protein ion charge states (Supplementary figure 2c; now denoted 'before optimized purification'). This reduces our ability to accurately deconvolute mass from these spectra - although upon re-analysis of the data we obtained a mass closer to the expected mass (342.7 kDa). To overcome this, we optimized our purification of AcrB to reduce artefactual LPS binding in accordance with: Reading, E. et al. The Effect of Detergent, Temperature, and Lipid on the Oligomeric State of MscL Constructs: Insights from Mass Spectrometry. Chem. Biol. 22, 593-603, doi:10.1016/j.chembiol.2015.04.016 (2015) – reference 45 in this manuscript.

This optimized purification regime produced well-resolved mass spectra with LPS removed, with some phospholipid binding maintained, which closely matches the expected mass for homotrimeric AcrB^{WT}-6xHis (Supplementary figure 2d). We have also included a mass spectra of AcrB^{G288D} in the Supplementary Figure 2e (using the same optimized purification protocol) with a measured mass corresponding to homotrimeric AcrB^{G288D}-6xHis. We have added that the G288D mutation did not affect the homotrimeric oligomeric state of AcrB in the main text also and present the new Supplementary Figure 2 and legend below:

Supplementary Figure 2. Biophysical characterization of AcrB^{WT} and AcrB^{G288D}. (a) SDS-PAGE of AcrB^{WT} and AcrB^{G288D} purified in DDM detergent micelles. (b) Circular dichroism and thermal melt (as determined by the loss of the helical signature at 222 nm) of AcrB within DDM detergent micelles reveals that the WT and G288D mutant have similar average secondary structure and thermal stability. (c) Native mass spectra of AcrB before removal of detectable bound lipopolysaccharides (LPS) using optimized purification procedure as previously described (see Methods). LPS binding to membrane proteins during purification has been previously identified by additional masses of 3.2–3.7 kDa from the apo form⁴⁵. Data was collected on a Synapt G2-Si mass spectrometer with AcrB^{WT} within Triton X-100 detergent micelles. (d) Native mass spectra of AcrB^{WT} within after optimized purification. Data was collected on a Thermo Scientific Q Exactive UHMR hybrid Quadrupole-orbitrap mass spectrometer with AcrB^{WT} within DDM detergent micelles. (e) Native mass spectra of AcrB^{G288D} after optimized purification. Data was collected on a Synapt G2-Si mass spectrometer with AcrB^{G288D} within Triton X-100 detergent micelles. (f) HDX-MS sequence coverage and peptide redundancy map of AcrB in DDM detergent micelles.

- I.138: “antibiotic binding only subtly alters the structural motions of AcrBWT”: how can the authors make sure that they have a stoichiometric binding and that the binding is not just transient...this sentence is possibly overstated or at least too general as the results might be different at different concentrations and for different antibiotics... The reviewer suggests to replace “antibiotics” by “CIP”.

The binding of different antibiotics could alter the structural dynamics of AcrB in distinct ways. We therefore agree with the reviewer that this statement is overstated - especially when considering the wide range of chemically distinct antibiotics which are substrates for AcrB efflux - and has been corrected.

AcrB has low affinity to a range of drugs; K_D values ranging from 5.5 to 74.1 μ M (Su, C. C. & Yu, E. W. Ligand-transporter interaction in the AcrB multidrug efflux pump determined by fluorescence polarization assay. FEBS Lett. 581, 4972-4976, doi:10.1016/j.febslet.2007.09.035 (2007) – reference 39 in this manuscript). HDX-MS has proven to be a robust method for obtaining both structural insights and dynamics information on both high affinity and low affinity protein-ligand interactions. The primary consideration when studying low-affinity protein-ligand interactions is to ensure high enough concentration of the partner protein or ligand is used under deuterium exchange conditions to ensure that all the target protein is bound in the complexed state. As the affinity of AcrB for CIP and PA β N are known (k_D of 74.1 and 15.7 μ M, respectively), we were able to initiate deuterium exchange by diluting an aqueous solution of AcrB in the presence of saturating concentrations of ligand in the equivalent buffer reconstituted in deuterium oxide (D_2O). The details for these experiments are provided in the “Preparation of ligands for hydrogen/deuterium mass spectrometry” section of the Methods and are in accordance with previous experimental design(s) for studying low affinity protein-ligand interactions by HDX-MS in: Chandramohan, A. et al. Predicting Allosteric Effects from Orthosteric Binding in Hsp90-Ligand Interactions: Implications for Fragment-Based Drug Design. PLOS Comput. Biol. 12, e1004840, doi:10.1371/journal.pcbi.1004840 (2016) – reference 49 in this manuscript.

- The sequence coverage seems different in the different conditions (CIP or PA β N), could the authors comment on that? On the same line, did the mutation induce a change in the sequence coverage of the region carrying the G288D mutation? Were the authors able to compare the peptide(s) carrying the mutation?

When drugs were added we saw a slight decrease in sequence coverage compared to apo AcrB (from 78.5 to 71.9%). This could be caused by the presence of small molecule drugs interfering with effective protein digestion, co-eluting with peptides adding to spectral complexity, and/or causing peptide ionisation suppression prior to infusion into the mass spectrometer. The differences observed between CIP and PA β N could also be related

to the reasons stated above. All metadata is presented in the Supporting Data Tables 1-2, which are provided with the manuscript.

We obtained good coverage of the G288D region (peptides 281-289, 281-290, 282-289) which allowed us to compare the data within $\Delta\text{HDX} = \text{AcrB}^{\text{G288D}} + \text{drug(s)} - \text{AcrB}^{\text{G288D}}$ (Supplementary Figure 11b), as the sequences were identical between the two states. Indeed, we found that PA6N led to reduced HDX within this region, which supports that it interacts with this region as shown with our MD simulations. To highlight this, we have provided the uptake plots for the 281-289, 281-290 and 282-289 peptides into a new Supplementary Figure 11b (shown below) and added a sentence to the main text section ‘PA6N EPI inhibits both wildtype and G288D AcrB’:

“Interactions with the aromatic residues of the HT involve hydrophobic stacking as well as cation- π attraction, not observed in AcrB^{WT}-PA6N, and possibly promoted by the direct interaction of the inhibitor with residue D288 (Supplementary Table 6; see Supplementary Discussion). This interaction is supported by the reduced HDX found within D288 containing peptides of AcrB^{G288D}-PA6N (Supplementary Figure 11b).”

We did not perform differential HDX (ΔHDX) on peptides containing G/D288 between AcrB^{WT} and AcrB^{G288D} because the peptide sequences are different (G vs. D), which can theoretically lead to distinct HDX uptake and/or LC retention behaviour based on the difference in peptide sequence alone and not by alteration in structural dynamics.

Supplementary Figure 11. Influence of drugs on AcrB^{G288D} structural dynamics. (a) Sum differential HDX (ΔHDX) plots for different drug conditions ($\Delta\text{HDX} = (\text{AcrB}^{\text{G288D}} + \text{drug(s)}) - \text{AcrB}^{\text{G288D}}$) for all time points collected. (b) Deuterium uptake plots for D288 containing peptides, 281-289, 281-290 and 282-289 for ($\text{AcrB}^{\text{G288D}} + \text{PA}\beta\text{N}$) – $\text{AcrB}^{\text{G288D}}$. All data reported as in Fig. 2 and 4. All HDX-MS peptide data can be found in Supporting Data Table 2.

- Simulations show the formation of hydrophobic bonds between PAβN and E130/K131/Q176/A619 (pose 1), E130/D276 (pose 2) and S126/S46/D174 (pose 3). How do the HDX-MS protection upon PAβN binding relate to these different proposed H-bonds? Are there any encompassing peptide significantly protected by PAβN? If yes this should be clearly stated in the text. If not, do the authors think that this might be due to a partial and/or transient binding of the drug? This point should be addressed in the manuscript.

The most stable pose of AcrB^{WT}-PAβN (Pose 1, discussed in the main text) is consistent with the experimental data: all residues involved in hydrogen bonds with PAβN are protected in the HDX-MS assay. Among the detected bonds (see Supplementary Table 3), several involve the switch loop or nearby residues, such as G619. These interactions may significantly contribute to the protection of the loop resulting from the HDX-MS data. Similarly, PAβN binding could be a factor in the protection of segments 129-137, 161-182 and 674-682. In all poses, indeed, residues belonging or adjacent to these segments (such

as E130, S133, Q176, L177 and E673) are involved in direct (and/or water-mediated) hydrogen bonds formed with the inhibitor. Additional residues, including S46 and D276, form stable hydrogen bonds with PA6N in at least one pose. Peptides adjacent to residue S46 (segment 26-44) are however unprotected according to HDX-MS data, while coverage on residue D276 is missing. Overall, a good agreement was detected between experimental data and intermolecular interactions between PA6N and the protein, indicating that stable contacts with the inhibitor may have a role in the protection of several regions of the DBP.

Analyses of the remaining systems also revealed a good correlation between the HDX-MS protection of specific peptides and substrate binding. In AcrB^{WT}-CIP-PA6N, indeed, residues belonging or adjacent to protected segments 138-149, 162-177 and 263-274 (such as 128-133, 174-176 and 273-276) are involved in hydrogen bonds or water-mediated interactions with the ligands in one or more poses (Supplementary Table 4). Similarly, in AcrB^{G288D}-PA6N, protection of peptides 291-300, 611-629 and 664-671 is coupled with intermolecular interactions involving the ligand. In this system, indeed, PA6N forms stable hydrogen bonds with residues D288, G616, F617 and I671 (Supplementary Table 6).

Following the reviewer's suggestion, we have inserted additional sentences, and/or re-worked paragraphs, in the main text of the paper to comment on the possible molecular interpretation of experimental data by means of MD simulations (see below). We have also provided additional discussions within the MD Supplementary Discussion and new associated Supplementary Tables:

- *In the "EPI restricts drug-binding pocket dynamics" section:*

"MD analysis of AcrB^{WT}-PA6N (T monomer) and apo AcrB^{WT} (L monomer) revealed that binding of PA6N is accompanied by an overall rigidification of the protein (Supplementary Fig. 7; see also Supplementary Discussion and Supplementary Tables 2-3), which involves large patches of the DBP, PBP, switch-loop, as well as the exit channel gate (EG), CH1, and CH2 channels. In particular: i) regions containing residues belonging/adjacent to the switch-loop that were found to directly interact with PA6N become more rigid in its presence, the extent of HDX protection upon PA6N binding (as revealed by the HDX-MS data) correlating with the formation of hydrogen bonds between the EPI and residues of (and nearby) the DBP; ii) the switch-loop itself (residues 615-620) features moderately enhanced hydration, whereas the nearby segments (residues 612-614 and 621-624) are overall dehydrated with respect to apo AcrB^{WT}. This supports an interaction between PA6N and the switch-loop region, which we anticipate being a key factor in mediating the mode of action of this EPI. More generally, the structural stabilisation that occurs upon PA6N binding might prevent local, as well as distal, functional movements that are key to substrate efflux along the transport pathway."

- ***In the “EPI and antibiotic can dually bind to AcrB” section:***

“The simultaneous binding of CIP and PA β N has similar effects on the flexibility and hydration of the DBP as the binding of PA β N only (Fig. 2a-b and Supplementary Fig. 10; see Supplementary Discussion). Several interactions contribute to stabilize this configuration, including the formation of hydrogen bonds between the substrates and several residues of the DBP (Supplementary Table 4) and stable intermolecular hydrogen bonds between the two ligands (Supplementary Fig. 9).”

- ***In the “PA β N EPI inhibits both wildtype and G288D AcrB” section:***

“Moreover, in accordance with HDX-MS, the switch-loop and the surrounding DBP region undergo further dehydration in AcrB^{G288D}-PA β N (Supplementary Fig. 13) and are involved in high-occurrence interactions with PA β N, similar to those formed within AcrB^{WT}-PA β N (Supplementary Tables 3 and 6).”

• Fig.2 shows the Δ HDX comparisons of (AcrBWT+CIP) vs. (AcrBWT) and (AcrBWT+PA β N) vs. (AcrBWT). We can clearly see that CIP mainly destabilizes the N-ter & C-ter of AcrBWT and stabilizes/protects the connecting-loop, while binding of PA β N additionally protects other regions, including the Switch loop. However, a proper statistical comparison between (AcrBWT+CIP) vs. (AcrBWT+PA β N) would be needed before stating that “However, in stark contrast to CIP, inhibitor binding led to HDX reduction throughout extensive parts of the PC1/PC2 cleft of the drug-binding pockets and within the connecting-loop (Fig.2a-b). This could signify inhibitor induced structural stabilization of the drug-binding pocket entrances” (l.139-140).

This has been performed as requested and is presented in the new Supplementary Figure 5 and is shown below. This new data comparison has enabled us to justify the above statement and we thank the reviewer for the suggestion. References to the figure have also been included into the main text where appropriate.

Supplementary Figure 5. Comparison of HDX-MS between AcrB^{WT} drug-binding conditions. Sum differential HDX (ΔHDX) plots for different drug conditions ($\Delta\text{HDX} = (\text{AcrB}^{\text{WT}} + \text{drug}(s)) - (\text{AcrB}^{\text{WT}} + \text{drug}(s))$) for all time points collected. All data reported as in Fig. 2. All HDX-MS peptide data can be found in Supporting Data Table 2.

- Fluorescence polarization binding and competition assays indicate that both molecule can interact independently with AcrB. This was further supported by MD simulations of AcrB in the Tstate with both CIP and PA β N nicely showing that PA β N is able to bind to the HT region while CIP lies next to the DBP-PBP interface. I just wonder how the CIP docking is affected by the additional presence of PA β N? Is CIP binding to the same site in the absence of PA β N? Could it be that the presence of PA β N stabilizes one particular bound state

of CIP, preventing its efflux? This seems to be suggested by the MD simulation analysis (supplementary material) but maybe not enough clearly stated in the main text.

We thank the reviewer for highlighting a point we overlooked in our original submission. Their suggestion prompted us to analyse in more detail the molecular features of the ternary complexes generated in silico. We also performed new calculations as detailed below.

The binding of CIP to AcrB^{WT} was previously investigated by computer simulations, which revealed that this compound could interact with the HT and with some residues of the DBP/PBP interface (see Fig. S6 and Table 1 in Vargiu & Nikaido, PNAS 109 (50), 20637). These results were compared with two additional MD simulations of AcrB^{WT}+CIP, performed following the protocol described the Methods section.

Importantly, both of AcrB^{WT}-CIP simulations were started from complex conformations where CIP is oriented differently with respect to the pose reported by Vargiu & Nikaido, PNAS 109 (50), 20637 (2012). These additional simulations confirmed the overall result reported previously, with CIP binding the HT independently of its orientation. The stable binding poses found in the present work for CIP in the presence of PA6N are different from those found in the binary complexes, due to the interaction of the inhibitor with a region of AcrB encompassing the binding site of CIP. As a result, the antibiotic is displaced towards an adjacent region of the DBP, where it establishes strong polar (hydrogen bonds) and hydrophobic interactions. Moreover, the two compounds interact also strongly with each other through hydrogen bonds between the carbonyl group of CIP and the amine and/or amide and/or guanidine groups of PA6N (see Supplementary Table 8). In the most stable pose (Pose 1, discussed in the main text) further stabilization is provided by stacking interactions between the quinolone group of CIP and the naphthyl moiety of the inhibitor. It is reasonable to expect that such an arrangement could further prevent a displacement of CIP towards the exit gate of AcrB.

Importantly, the simultaneous binding of PA6N and CIP within this region of AcrB occurs in the absence of large rearrangements of the transporter (as compared to the available X-ray structures (see Supplementary Table 9), reinforcing the possibility of simultaneous binding of two compounds at and around the DBP of AcrB.

We have included detail on the outcome of these comparisons in the Supplementary Discussion and provided a new Supplementary Figure 16 with the poses for AcrB^{WT}-CIP and AcrB^{G288D}-CIP (shown below).

Supplementary Figure 16. Representative binding poses of AcrB^{WT}-CIP and AcrB^{G288D}-CIP (upper panel) and high-occupancy hydrogen bonds (H-bonds) established between CIP and the protein (lower table). See Supplementary Fig. 6 for further details on the representation of the binding poses. As stated in the Supplementary Methods, the binding pose of AcrB^{WT}-CIP features the substrate within the same region as in Ref. [Vargiu and Nikaido, PNAS, 2012], although the orientation is opposite. Hydrogen bond analyses were performed on the last 300 ns of each simulation (see Methods). Only direct or water-mediated bonds with occupancy higher than 20% have been reported.

- The authors then present in Fig.2 the Δ HDX comparison of (AcrB^{WT}+CIP+PA β N) vs. (AcrB^{WT}), which indeed shows very similar behavior as the previous comparison (AcrB^{WT}+PA β N) vs. (AcrB^{WT}), suggesting that most fluctuations are due to PA β N and that there isn't much synergy between the substrate and the EPI. However, to see the additional effect of each molecule, identify which regions are statistically altered synergistically and reach these conclusions, proper statistical comparisons between the states (AcrB^{WT}+CIP+PA β N) vs. (AcrB^{WT}+CIP) and (AcrB^{WT}+CIP+ PA β N) vs. (AcrB^{WT}+PA β N) are required (at least in the supplementary material).

The requested differential HDX (Δ HDX) plots have been created and are presented in the new Supplementary Figure 5. The new plots show that AcrB^{WT}+CIP+PA β N largely acts like AcrB^{WT}+PA β N. This has strengthened our conclusions and we thank the reviewer for the suggestion for its inclusion. A reference to the figure has also been included into the main text where appropriate.

- The local and allosteric effects of the G288D mutation on the dynamics of AcrB are quite impressive, as shown by the changes in HDX observed on Fig.4b (no drugs). Irrespective

of the drug(s), the G288D mutation induces local flexibility (increase HDX) and a long-range stabilization (decrease HDX) of the connecting loop, as seen in Fig.4c. The effects of the drugs are first represented by comparing the HDX of AcrBG288D+CIP or PAβN with AcrBWT, meaning that two parameters (mutation and presence of the drug) are changing and that the contribution of each parameter is hard to apprehend. Actually, it seems to me that the local destabilization of the PN2 region is only due to the G288D mutation rather than by the drugs as it seems to be the case in Fig.4c. The authors then rightfully compare (AcrBG288D+CIP) vs. (AcrBG288D) and (AcrBG288D+PAβN) vs. (AcrBG288D) (supplementary Fig.9), which is better, I think to see the sole effect of each drug on the mutant. In this representation, the destabilization of the PN2 region by the drugs is not visible anymore. I would thus recommend to present these comparisons (AcrBG288D+drug) vs. (AcrBG288D) in the main text and the one from Fig.4c in the supplementary data (just keeping the AcrBG288D vs. AcrBWT comparison from Fig.4c), in order to avoid any misleading interpretation of the data. Alternatively, the authors could represent the comparison of (AcrBG288D+drug) vs. (AcrBWT+drugs) to see the effect of the mutation on the drug-bound AcrB (as mentioned later on lines 295-302).

The reviewer makes many good points. Regrettably, we previously annotated the original figure with:

- $\Delta\text{HDX} = (\text{AcrB}^{\text{G288D}} \text{ +/- drug(s)}) \text{ vs. } (\text{AcrB}^{\text{WT}} \text{ +/- drug(s)})$

Which was misleading, and we apologize for this. Due to this lack of clarity within Fig. 4 we have caused confusion with what data is, and is not, presented in Fig. 4. In Fig. 4b we present both (AcrB^{G288D} vs. AcrB^{WT} – both apo forms) and ((AcrB^{G288D}+drug) vs. (AcrB^{WT}+drugs) – both holo forms), as was recommended by the reviewer. We have now corrected the labelling within Fig. 4b, annotating the top panel as:

- $\Delta\text{HDX} = \text{AcrB}^{\text{G288D}} - \text{AcrB}^{\text{WT}}$

and the bottom three panels as:

- $\Delta\text{HDX} = (\text{AcrB}^{\text{G288D}} + \text{drug(s)}) - (\text{AcrB}^{\text{WT}} + \text{drug(s)})$

Indeed, we agree with the reviewer that this allows us to best monitor the effect of the G288D mutation on AcrB structural dynamics; enabling us to decipher the substrate-independent effect of the mutant on the HDX behaviour – which is highlighted on the structure in Fig. 4c and within the dashed boxes in Fig. 4b.

To better clarify our data and our conclusions we have also included re-worked discussions of the $\Delta\text{HDX} = (\text{AcrB}^{\text{G288D}} + \text{drug(s)}) - (\text{AcrB}^{\text{G288D}})$ results, which is presented in

Supplementary Figure 11a, within the “MDR-conferring substitution, G288D, affects drug-binding pocket dynamics”:

“Upon substrate binding, PA6N caused reduced HDX within the PC1/PC2 regions for AcrB^{G288D} (Supplementary Figure 11), as was observed for AcrB^{WT} (Fig. 2a), with AcrB^{G288D}-PA6N having increased HDX reduction within PC1 and R2 (TM 7-12) domains in comparison to AcrB^{WT}-PA6N (Fig. 4b). This supports that the PA6N EPI affects the dynamics of the different AcrB genotypes in a similar manner, with AcrB^{G288D}-PA6N possibly undergoing further restraint than AcrB^{WT}-PA6N. Where CIP caused increased HDX throughout extensive regions of AcrB^{G288D} (Supplementary Figure 11), bringing the PC1/PC2 and DC regions of AcrB^{G288D} closer in parity with AcrB^{WT} for both CIP and CIP-PA6N conditions (Fig. 4b).

Markedly, in the apo form and for all three substrate conditions tested (CIP, PA6N, and CIP-PA6N), the G288D substitution consistently caused increased HDX within the PN2 region and decreased HDX of the connecting-loop (Fig. 4b). Signifying that these effects are retained even upon substrate binding and, due to their close structural proximity, may relate to concerted changes to the dynamics of the substrate translocation pathway (Fig. 4c).”

- The comparison throughout the manuscript of the first hydration shell and HDX levels is very interesting. The authors claim several times that these results of simulations are in good agreement with their HDX data. I think it would be a good idea, for a non-expert readership, to detail a little bit more in what sense these results agree for eg. by saying (in the main text) that a reduced hydration shell should imply reduced HDX and why.

We agree with the reviewer that further detail would aid a broad readership in understanding our data and conclusions. We have therefore provided more detail in the “EPI restricts drug-binding pocket dynamics” section to better explain the above point:

“To further understand the dynamics of AcrB in the presence of substrates, Root Mean Square Fluctuation (RMSF) and first hydration shell profiles were compared to the HDX-MS data. The average number of water molecules in the first amide NH solvation shell computed by MD simulations has been found to correlate well with HDX³⁷. A reduced hydration shell should therefore imply reduced HDX, due to the decrease in specific interactions between amide N-H bonds and the solvent. However, protein HDX is complex, with neighbouring residues having significant differences in their solvent interactions, this combined with the stark contrast between MD simulation and HDX-MS experimental time scales (μ s to ms versus seconds to hours) means that a simple quantitative comparison can often be incomplete. Nevertheless, comparisons to MD calculated hydration profiles can provide informative qualitative interpretation of protein HDX.”

With a new reference added also:

- 37 Petruk, A. A., Defelipe, L. A., Rodríguez Limardo, R. G., Bucci, H., Marti, M. A. & Turjanski, A. G. *Molecular Dynamics Simulations Provide Atomistic Insight into Hydrogen Exchange Mass Spectrometry Experiments. J. Chem. Theory Comput.* 9, 658-669, doi: 10.1021/ct300519v (2013).

Further to this we have also reworded our introduction to HDX-MS within the ‘Hydrogen/Deuterium eXchange Mass Spectrometry of AcrB’ section to aid the understanding of the HDX process, and what this measures, for a non-expert readership:

“HDX-MS is a solution-based method which can provide molecular level information on local protein structure, stability, and dynamics³¹⁻³³. HDX occurs when backbone amide hydrogens are made accessible to D₂O solvent through structure unfolding and H-bond breakage; HDX is fast within unfolded regions and slow within stably folded regions (i.e. α -helices, β -sheet interiors), where local structural fluctuations which expose an otherwise protected amide hydrogen to solvent transiently are required for HDX to occur.”

• On the same line, the fact that hydrogen bonds were found in the MD ternary complex simulations, between the two drugs (and not only with the HT and switch-loop) is, I think, important to state in the main text.

We thank the reviewer for their suggestion. Following which, we added a few sentences to the main text of the paper to highlight the presence of such interactions:

- **In the ‘EPI and antibiotic can dually bind to AcrB’ section:**

“Several interactions contribute to stabilize this configuration, including the formation of hydrogen bonds between the substrates and several residues of the DBP (Supplementary Table 4) and stable intermolecular hydrogen bonds between the two ligands (Supplementary Fig. 9).”

- **In the ‘PA β N EPI inhibits both wildtype and G288D AcrB’ section:**

“Similar conclusions emerged from the comparison between AcrB^{G288D}-CIP-PA β N and AcrB^{WT}-CIP-PA β N. Indeed, MD simulations of the former complex revealed that, even upon G288D substitution, CIP and PA β N can stably occupy the DBP at the same time (Fig. 5a). As in AcrB^{WT}-CIP-PA β N, stabilizing interactions include several substrate contacts with the AcrB^{G288D} protein, also involving D288 (Supplementary Fig. 14-15 and Supplementary

Table 7), as well as intermolecular hydrogen bonds between the two ligands (Supplementary Table 8)."

- The antibiotics used for MIC assays are different in Fig.2c and Fig.5c (apart from CIP), could the authors explain why?

The MIC data presented in Fig.2c was to show the effect of increasing PABN concentration on the MIC of three different classes of antibiotics (CIP, TET, and CHL), all known AcrB substrates, in a wildtype E. coli with wildtype AcrB.

In Fig.5b (now Fig. 5d) we focused on substrates such as CIP and MIN as they are both well-known and because AcrB^{G288D}, discovered within Salmonella clinical isolates previously, was found to have increased and decreased susceptibility to MIN and CIP respectively. By studying these AcrB substrates in the presence of the inhibitor PAβN we planned to observe what, if any, effect we would see with the different AcrB genotypes (AcrB-WT & AcrB-G288D). Further clarification of this choice has been included in the main text, within the 'PAβN EPI inhibits both wildtype and G288D AcrB' section:

"These findings were supported by bacterial susceptibility assays on E. coli containing overexpressed AcrB^{G288D}. AcrB^{G288D} was previously discovered within Salmonella clinical isolates³⁰ and found to have increased and decreased susceptibility to MIN and CIP antibiotics, respectively. We chose, therefore, to study these AcrB substrates in the presence of the inhibitor PAβN to observe what, if any, effect would be seen with the different AcrB genotypes. PAβN incubation led to increased MIN and CIP antibiotic susceptibility for both AcrB^{WT} and AcrB^{G288D} (Fig. 5d). AcrB^{G288D} being more susceptible to PAβN than AcrB^{WT}. The decreased susceptibility of AcrB^{G288D} to CIP, found in Salmonella³⁰, was not recapitulated in our assays using the laboratory E. coli strain MG1655. This may be due to CIP efflux via another transporter found in E. coli but not in Salmonella. However, the associated increased susceptibility to MIN was observed (Fig. 5d), supporting that G288D has a profound impact on AcrB substrate efflux within both E. coli and Salmonella."

- The last sentence of the results states that "G288D has a profound impact on AcrB substrate specificity within both E. coli and Salmonella", however the MD simulations clearly showed that both WT and G288D AcrB were able to bind to both CIP (with and w/o PAβN). I don't clearly see how the data presented here infers directly a difference in substrate "specificity" per se (in terms of Kd for eg.) but rather on substrate export (indirectly with the MIC assays) or stability (HDX and MD simulations). Could the authors use fluorescence polarization to determine and compare the Kds of CIP and PAβN to AcrBG288D as they did for the WT construct?

The reviewer makes a very good point and suggestion for further experiments. We have performed the analogous fluorescence polarization binding and competition assays used for AcrB^{WT} on AcrB^{G288D}. We found parity between the binding strengths of CIP to AcrB^{G288D}-PABN and AcrB^{WT}-PABN and observed that PABN could not outcompete CIP binding to both AcrB^{G288D} and AcrB^{WT}. Although AcrB^{G288D}-PABN had a slightly higher binding affinity to CIP than AcrB^{WT}-PABN (K_D of $67.3 \pm 13.2 \mu\text{M}$ (AcrB^{WT}) versus $22.3 \pm 3.1 \mu\text{M}$ (AcrB^{G288D})). We argue that this demonstrates that the binding specificity and inhibitory action is similar between AcrB^{G288D} and AcrB^{WT}. At least for the CIP-PABN system investigated here. This supports the reviewer's statement that we are not directly observing altered specificity but rather differences in efflux and stability.

Indeed, we consistently argue within the manuscript that the alteration in structural dynamics/stability explains why we observe differences in efflux; therefore, we agree with the reviewer that statements on specificity need to be changed. We have therefore changed the reference to "altered substrate specificity" throughout the manuscript and replaced with "altered substrate efflux", this has been changed within the below locations.:

- Title: from "Perturbed structural dynamics underlie inhibition and altered specificity of the multidrug efflux pump AcrB" to "Perturbed structural dynamics underlie inhibition and altered efflux of the multidrug pump AcrB".
- Abstract: from "altered substrate specificity" to "altered substrate efflux".
- Introduction: from "substrate specificity" to "substrate efflux".
- Results: from "substrate specificity" to "substrate efflux".
- Discussion: from "altered substrate specificity" to "altered substrate efflux".

The new data figure and corresponding text have also been added to manuscript within the "PA6N EPI inhibits both wildtype and G288D AcrB" section:

"Fluorescence polarization binding and competition assays support that a ternary AcrB^{G288D}-CIP-PA6N is also possible (Fig. 5b-c): i) CIP binds to a preformed AcrB^{G288D}-PA6N complex (K_D of $22.7 \pm 2.9 \mu\text{M}$) with similar, albeit slightly higher, affinity compared to CIP binding to AcrB^{WT}-PA6N (K_D of $67.3 \pm 13.2 \mu\text{M}$); ii) titration of the PA6N inhibitor could not effectively outcompete CIP binding from AcrB^{G288D}-CIP, as was found for AcrB^{WT}-CIP (Fig. 3b). Taken together, the fluorescence polarization binding assays and MD simulation data advocate that AcrB^{G288D} is inhibited by PA6N in a similar manner as AcrB^{WT}."

Figure 5. *AcrB*^{G288D} is inhibited by the EPI PAβN. (a) Molecular docking and multi-copy μs-long MD simulations reveal stable interactions of CIP (orange) and PAβN (cyan) to *AcrB*^{G288D} T-state monomer and show their likely binding locations. The pose and its orientation are the same as shown for *AcrB*^{WT} in Fig. 3c. EG = exit channel gate (blue spheres), SL = switch-loop (yellow), and HT = hydrophobic trap (purple). All computational data, including binding free energies can be found in Supplementary Table 5 and Supplementary Fig. 8, 14-16. (b) Binding of CIP by *AcrB*^{G288D} in the presence of 150 μM of PAβN as determined by a fluorescence polarization assay performed by Su et al.³⁹. All data are reported and fit as in Fig. 3a ($R^2 = 0.99$). (c) Binding competition assay between PAβN and CIP for *AcrB*^{WT}. All data are reported as in Fig. 3b. (d) MIC assays of *Escherichia coli* containing *AcrB*^{WT} or *AcrB*^{G288D} in the presence of PAβN and antibiotics. *AcrB* was overexpressed in MG1655 Δ*acrB* from a pBR322 plasmid containing its corresponding *acrAB* genes, natural promoter and ‘marbox’ sequence. Minocycline = MIN, Ciprofloxacin = CIP, and phenylalanine-arginine-β-naphthylamide = PAβN. †PAβN was added at a concentration of 50 μg/ml.

We have also included the zero *AcrB* concentration point for the CIP binding data to *AcrB*^{WT}-PABN, which was not included in the previous manuscript draft in Fig. 3a. Inclusion of this data point slightly improved the accuracy of our fit and K_D measurement (K_D of $76.2 \pm 17.4 \mu\text{M}$ to K_D of $67.3 \pm 13.2 \mu\text{M}$) and, although the number was altered slightly, it was still insignificantly different to the K_D found by Su et al.³⁹ for CIP binding to *AcrB*^{WT} and, therefore, does not change any of our previous conclusions. The new figure and K_D value have been added to manuscript within the “EPI and antibiotic can dually bind to *AcrB*” section:

Figure 3. Dual binding of Ciprofloxacin antibiotic and PAβN inhibitor to the DBP of AcrB^{WT}. (a) Binding of CIP by AcrB^{WT} in the presence of 150 μM of PAβN as determined by a fluorescence polarization assay performed by Su et al.³⁸. CIP was maintained at 1.5 μM throughout and its emission wavelength measured at 415 nm. Reported data are the average and standard deviation from repeated measurements (n = 3) and were fitted to a hyperbola function ($FP = (B_{max} * [protein]) / (K_D + [protein])$, $R^2 = 0.98$). (b) Binding competition assay between PAβN and CIP for AcrB^{WT}. PAβN was non-fluorescent in the experimental conditions. Data are the average and standard deviation from repeated measurements (n = 3). (c) Molecular docking and multi-copy μs-long MD simulations reveal stable interactions of CIP (orange) and PAβN (cyan) to AcrB^{WT} T-state monomer and show their likely binding locations. EG = exit channel gate (blue spheres), SL = switch-loop (yellow), and HT = hydrophobic trap (purple). All computational data can be found in Supplementary Table 2 and Supplementary Fig. 8-9.

- The authors use a ~1000:1 ligand to AcrB ratio for the HDX experiments: how to make sure that in these conditions, non-specific binding does not occur? This could obviously affect the regions pinpointed by the HDX experiments. Do the authors expect to see a single drug bound to AcrB under such conditions in native MS?

We chose these ligands:protein ratios to ensure close to complete binding to the target protein under our deuterium exchange conditions based on the known dissociation constants of the drugs, which are in the μM range, in line with previously published material (Chandramohan, A. et al. Predicting Allosteric Effects from Orthosteric Binding in Hsp90-Ligand Interactions: Implications for Fragment-Based Drug Design. PLOS Comput. Biol. 12, e1004840, doi:10.1371/journal.pcbi.1004840 (2016)). In addition, blind molecular docking

clearly shows that binding of CIP and PABN to the DBP and/or PBP of AcrB occurs with the highest frequency and is associated with the highest score (Supplementary Figure 8 and Supplementary Tables 2-3). Furthermore, state-of-the-art MD simulations confirm the formation of stable binary and ternary complexes between AcrB and CIP, PABN or PABN+CIP. We, therefore, expect most binding events to be within the drug-binding pockets of AcrB.

*To investigate the binding stoichiometry, we attempted to collect Native MS data of AcrB bound to the drugs, but we were unable to monitor the AcrB-drug complex in the gas-phase. This is likely due to the high activation (collision voltage) energies required to strip DDM detergent from membrane proteins before MS detection (see: Reading, E. et al. The role of the detergent micelle in preserving the structure of membrane proteins in the gas phase. *Angew. Chem. Int. Ed Engl.* 54, 4577-4581, doi:10.1002/anie.201411622 (2015) – reference 44 in the manuscript) causing stripping of the low affinity drug molecules. Additionally, the requirement for proteins to be present in volatile ammonium acetate solutions for efficient nano-electrospray could also affect noncovalent protein-ligand formation.*

- The HDX-MS data was made available through on Pride (#416950) but I could not find it. The authors need to make sure that the data is either publicly available or provide login and password to the reviewers. The deposited data should include .raw files, DynamX .csv output files, .fasta files and PLGS output files.

Our dataset "HDX-MS reveals the perturbed structural dynamics underlie inhibition and altered specificity of the multidrug efflux pump AcrB" has been successfully submitted to ProteomeXchange via the PRIDE database. The data is currently private and can be accessed a single reviewer account that has been created. We have been informed by PRIDE that it is essential we notify them after the first (online) publication of the corresponding manuscript. Otherwise the data will remain inaccessible to readers.

For access to the account by the reviewer during the peer review process the Reviewer account details are provided below:

Project accession: PXD019047

Reviewer account details:

Username: reviewer33524@ebi.ac.uk

Password: 961rnZhP

Additionally, we have added the following sentence into the "Hydrogen/deuterium mass spectrometry" section of the Methods, as recommended by PRIDE:

"The mass spectrometry proteomics data have been deposited to the ProteomeXchange Consortium via the PRIDE partner repository (<https://www.ebi.ac.uk/pride/>) with the dataset identifier PXD019047".

Minor Comments:

- Supplementary Fig.2e: please map the TM domains to the sequence.

This has been corrected but reassigned as Supplementary Fig. 2f.

- Supplementary Fig.3 shows the HDX heatmap obtained for AcrB^{WT} alone. The reviewer suggests to include (panel C) a 3D structure color-coded with this HDX-MS data (last time point for eg.).

We have generated an additional Supplementary Figure 4, which displays the HDX-MS heat maps from Supplementary Figure 3 translated onto the 3D structure of AcrB^{WT}; this has been cited in the main text where appropriate.

- l.283: "how the mutation effects" should read "how the mutation affects".

This has been corrected within the text.

- The top legend of Supplementary Fig.9 should read (AcrBG288D+drug(s)) instead of AcrBG288D+/-drug(s).

This has been corrected but has now been reassigned to Supplementary Fig. 11.

In the Material and Methods section:

- the flow rates for the protein purification (elution on HisTrap and SEC) should be indicated.

A sentence has been added to the 'AcrB overexpression and purification' Methods section:

"A flow rate of 1 ml/min was used during HiTrap and SEC purification."

- the MWCO of the spin column for the concentration step should be indicated.

The below sentence has been modified within the 'AcrB overexpression and purification' Methods section:

"Peak fractions eluted from the SEC column containing pure AcrB were pooled, spin concentrated using a 100K MWCO concentrator (Amicon®), and spin filtered before being flash frozen and stored at -80 °C."

- the argon trap collision gas flow rate should be indicated.

A sentence has been added to the ‘Hydrogen/deuterium mass spectrometry’ Methods section:

“Argon was used as the trap collision gas at a flow rate of 2 mL/min.”

- l.670: “Bipharma” should read “Biopharma”.

This has been corrected within the text.

• HDX: the volume of sample transferred to the quench solution, the volume of quench solution and the volume of sample injected (as well as the volume of the loop) should be indicated.

This has been corrected within the ‘Hydrogen/deuterium mass spectrometry’ Methods section.

Finally, we thank the reviewers for their careful reading and constructive comments and hope that the manuscript is now suitable for publication in *Nature Communications*.

REVIEWERS' COMMENTS

Reviewer #1 (Remarks to the Author):

The authors have addressed all my questions and the manuscript is now ready for publication.

Reviewer #2 (Remarks to the Author):

The revised manuscript is much improved, and has addressed adequately my principal concerns. I'm pleased to see the use of effect and affect rather than impact. I note, however, in line 99, the proper word is "affecting" rather than "effecting". :)

Reviewer #3 (Remarks to the Author):

The authors have made notable efforts to successfully and thoroughly address all the major and minor issues I had on the previous manuscript. I now recommend it for publication in Nature Communications.

Julien Marcoux

REVIEWERS' COMMENTS (2nd revision)

Reviewer #1 (Remarks to the Author):

The authors have addressed all my questions and the manuscript is now ready for publication.

We thank the reviewer for their assessment of our revised manuscript.

Reviewer #2 (Remarks to the Author):

The revised manuscript is much improved, and has addressed adequately my principal concerns. I'm pleased to see the use of effect and affect rather than impact. I note, however, in line 99, the proper word is "affecting" rather than "effecting". :)

We thank the reviewer for their assessment of our revised manuscript. We have now performed the recommended correction.

Reviewer #3 (Remarks to the Author):

The authors have made notable efforts to successfully and thoroughly address all the major and minor issues I had on the previous manuscript. I now recommend it for publication in Nature Communications.

Julien Marcoux

We thank the reviewer for their assessment of our revised manuscript.